# ThinkDial: An Open Recipe for Controlling Reasoning Effort in Large Language Models

## Abstract

Large language models (LLMs) with chain-of-thought reasoning have demonstrated remarkable problem-solving capabilities, but controlling their computational effort remains a significant challenge for practical deployment. Recent proprietary systems like OpenAI's gpt-oss series have introduced discrete operational modes for intuitive reasoning control, but the open-source community has largely failed to achieve such capabilities. In this paper, we introduce ThinkDial, the first open-recipe end-to-end framework that successfully implements gpt-oss-style controllable reasoning through discrete operational modes. Our system enables seamless switching between three distinct reasoning regimes: High mode (full reasoning capability), Medium mode (50% token reduction with ≤10% performance degradation), and Low mode (75% token reduction with ≤15% performance degradation). We achieve this through an end-to-end training paradigm that integrates budget-mode control throughout the entire pipeline: budget-mode supervised fine-tuning that embeds controllable reasoning capabilities directly into the learning process, and two-phase budget-aware reinforcement learning with adaptive reward shaping. Extensive experiments demonstrate that ThinkDial achieves target compression-performance trade-offs with clear response length reductions while maintaining performance thresholds. The framework also exhibits strong generalization capabilities on out-of-distribution tasks.

## 1 Introduction

The advancement of large language models (LLMs) has led to remarkable capabilities in complex reasoning tasks through extended reasoning chains (Sui et al., 2025; Feng et al., 2025). However, these models often generate unnecessarily lengthy reasoning processes with redundant steps and circular reasoning, leading to increased computational costs, potential quality degradation through error propagation, and reduced interpretability (Sui et al., 2025; Feng et al., 2025). This poses a significant challenge for practical deployment, where different scenarios may require different levels of reasoning depth and computational budget.

Recent breakthrough systems, notably OpenAI's gpt-oss series (OpenAI, 2025), have demonstrated remarkable reasoning capabilities while introducing an innovative paradigm for controlling computational effort through discrete operational modes (e.g., "Low", "Medium", "High"). Unlike explicit token budget methods that require users to specify exact computational constraints (Anthropic, 2025), the gpt-oss paradigm provides: *(1) User accessibility*: users can specify their preference without needing technical knowledge of token economics; *(2) Dynamic allocation*: the system can dynamically allocate computational resources based on problem complexity rather than rigid constraints; Unlike adaptive CoT approaches that are limited to binary switching between thinking and non-thinking modes (Lou et al., 2025), the gpt-oss framework enables: *(3) Fine-grained control*: multiple efficiency priorities (Low, Medium, High) can be applied to the same problem based on user requirements.

Despite the clear superiority of this mode-based paradigm, the open-source community has largely failed to achieve such capabilities. Existing controllable generation approaches predominantly require explicit specification of token budgets (Muennighoff et al., 2025; Sun et al., 2025; Aggarwal & Welleck, 2025; Anthropic, 2025) or rely on adaptive switching between thinking and non-thinking modes (Lou et al., 2025; Zhang et al., 2025a; Bai et al., 2023), both of which lack the intuitive three-

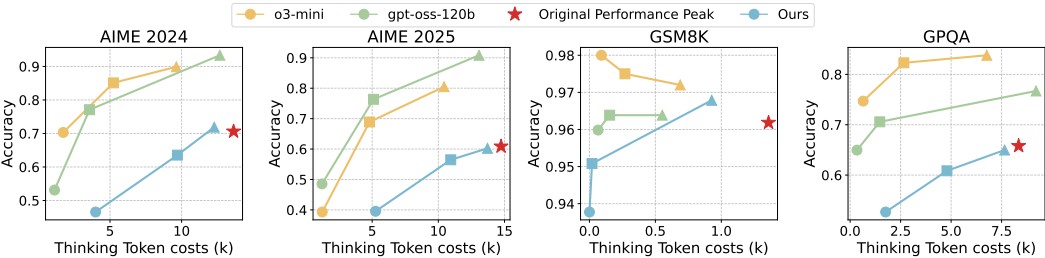

Figure 1: Comparison of our THINKDIAL and gpt-oss-style model in controllable reasoning. The red star indicates the performance ceiling achievable by the Qwen2.5-32B-Instruct model after RL training. Circles, squares, and triangles represent Low, Medium, and High modes, respectively.

mode control paradigm. This gap represents a significant limitation in democratizing advanced reasoning capabilities and has created an urgent need for open-recipe solutions that can match the sophistication of proprietary systems.

In this paper, we introduce THINKDIAL, the first open-recipe end-to-end framework to successfully implement gpt-oss-style controllable reasoning through discrete operational modes. As shown in Figure 1, our system enables seamless switching between three distinct reasoning regimes: *High mode* (full reasoning capability), *Medium mode* (50% token reduction with ≤10% performance degradation), and *Low mode* (75% token reduction with ≤15% performance degradation).

The technical foundation of our approach rests on an end-to-end training paradigm that integrates budget-mode control throughout the entire pipeline: *(1) Budget-Mode Supervised Fine-tuning*: Rather than retrofitting existing models with RL compression techniques, we embed controllable reasoning capabilities directly into the SFT learning process. The key insights are: *(a)* models must learn to naturally associate different mode specifications with appropriate reasoning patterns, and *(b)* we must first establish stable output distributions for each mode to prevent interference between different modes during RL training. *(2) Budget-Aware Reinforcement Learning*: To enable the model to seamlessly switch between three budget modes while preserving its performance ceiling, we employ a two-phase RL training strategy: first conducting warm-up RL training to reach optimal performance, then implementing budget-aware reward shaping with different length rewards for each mode, allowing the model to learn distinct reasoning capabilities with varying response lengths across different modes without compromising its peak reasoning ability. Additionally, we discover that models increasingly exhibit reasoning leakage from thinking sections to answer sections during RL training, a phenomenon we term "Reasoning Length Hacking". This behavior stems from aggressive compression in Low mode SFT data that caused reasoning overflow, which RL training inadvertently reinforces. Thus, we incorporate Leak Penalty in our reward shaping strategy.

Extensive experiments across multiple mathematical reasoning benchmarks demonstrate that THINKDIAL successfully achieves the target compression-performance trade-offs with remarkable consistency. Our results show clear step-wise thinking token reductions (High → Medium → Low) while maintaining the specified performance thresholds across diverse problem types. The framework also exhibits strong generalization capabilities, maintaining controllable behavior even on out-of-distribution tasks, despite being trained primarily on mathematical reasoning data.

The main contributions of this work include: *(1)* The first open-recipe implementation of reasoning control, moving beyond token budget specification. *(2)* An end-to-end training framework that integrates budget-mode control from supervised fine-tuning through reinforcement learning, featuring adaptive reward shaping. *(3)* Comprehensive experimental validation demonstrating robust controllable reasoning across multiple benchmarks with strong out-of-distribution generalization.

## 2 RELATED WORK

**CoT Compression.** Large reasoning models suffer from overthinking, where models generate excessively lengthy chains with redundant steps, circular reasoning, or unnecessary elaboration (Sui et al., 2025; Feng et al., 2025), leading to increased computational costs, potential quality degrada-

tion through error propagation, reduced interpretability, and inefficient resource utilization. Many existing works have addressed the overthinking problem through CoT compression techniques, including: (1) constructing supervised fine-tuning (SFT) datasets by selecting short yet correct reasoning chains (Xia et al., 2025; Ma et al., 2025; Kang et al., 2025) and learning summary tokens for training (Zhang et al., 2025b), (2) reward shaping during reinforcement learning that rewards short and correct answers (Shen et al., 2025; Team et al., 2025; Luo et al., 2025), provides dense rewards for higher-quality reasoning processes (Qu et al., 2025), or rewards early-exit rollouts that stop sufficiently early yet can be resumed to the correct answer (Dai et al., 2025), and (3) inference-time scaling methods that explore higher-quality intermediate reasoning processes to achieve CoT compression (Lu et al., 2025). However, these methods only compress CoT and do not provide controllable reasoning, i.e., allowing users to adjust the reasoning depth as needed.

**Control Reasoning Effort.** Several works have investigated how to enable models to achieve controllable prioritization between efficiency and performance. (1) *Explicit Reasoning Token Budget:* Some methods use parameters to control reasoning length by setting explicit token budgets (Muennighoff et al., 2025; Sun et al., 2025; Aggarwal & Welleck, 2025; Anthropic, 2025). However, it is challenging to determine appropriate budgets for problems of varying difficulty, such as comparing GSM8K and AIME problems. (2) *Adaptive CoT:* These approaches allow models to autonomously decide between thinking and non-thinking modes based on the query (Lou et al., 2025; Zhang et al., 2025a; Bai et al., 2023). However, they are limited in their ability to provide multiple efficiency priorities for the same problem. (3) *Three-Mode Systems* offer a structured compromise by providing discrete reasoning levels (low, medium, high) that enable users to explicitly specify their preference for the efficiency-performance trade-off (OpenAI, 2025): prioritizing speed for time-sensitive applications, emphasizing accuracy for critical decisions, or balancing both for general use cases. This approach provides intuitive control without requiring users to understand token budgets. However, existing three-mode implementations remain closed.

Compare to previous work, we provide the first open recipe that unifies multiple controllable reasoning regimes within a single model that can switch modes via lightweight system prompts. THINKDIAL delivers stable, discrete, and user-level controllability across Low, Medium and High budgets, and the design naturally generalizes to more than three modes. This fills a critical gap in open-source controllable reasoning and offers a reproducible framework that matches the sophistication previously available only in proprietary systems.

## 3 METHOD

We propose an end-to-end training paradigm for models to learn controllable reasoning capabilities through three sequential steps: *(1) Budget-Mode Supervised Fine-tuning* to embed mode-specific reasoning patterns directly into the model's base capabilities; *(2) Stage-1 RL training* to establish peak performance foundation, thus the following RL training will not compromise the model's peak reasoning ability; *(3) Stage-2 RL training* to implement budget-aware reward shaping that enables seamless switching between reasoning modes.

### 3.1 BUDGET-MODE SUPERVISED FINE-TUNING

Previous approaches to controllable reasoning typically focus on the RL training phases. However, we argue that the SFT stage is crucial for establishing the foundation of controllable reasoning capabilities for two reasons: *(a)* models must learn to naturally associate different mode specifications with appropriate reasoning patterns, and *(b)* we must first establish stable output distributions for each mode to prevent interference between different modes during RL training.

We construct specialized training data that demonstrates how the same problem can be solved with different levels of reasoning depth while maintaining correctness. As shown in Figure 2, to enable the switching of model output distributions across different modes, we design distinct system prompts for each budget mode. To construct solutions of different budget mode, we start from full chain-of-thought solutions (High mode) and derive Medium and Low variants by truncating the thinking process to approximately $r_{medium}=50\%$ and $r_{low}=25\%$ of the original length. Truncation is applied only at $\backslash n \backslash n$ step boundaries so that every retained segment corresponds to a complete reasoning step. After truncation, we append a lightweight mode-specific connective phrase before

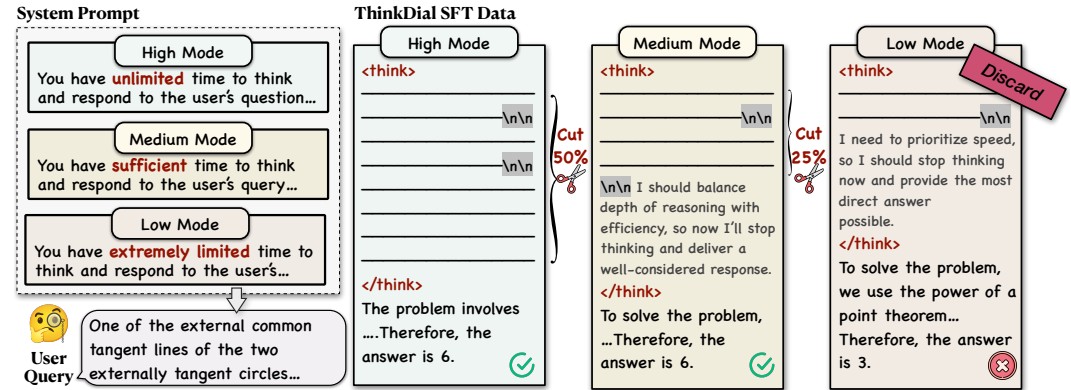

Figure 2: The illustration of data construction for budget-mode supervised fine-tuning.

the `</think>` marker to maintain coherence, and regenerate the final answer to ensure correctness under shorter reasoning contexts. All candidates are verified and only those with correct final answers are kept. This process yields parallel high, medium, and low reasoning trajectories for the same problem, enabling the model to learn stable budget-aware reasoning behaviors. The detailed construction details, and corresponding examples of the SFT data can be found in Appendix D.

The SFT training objective minimizes negative log-likelihood loss across all modes:

$$\mathcal{L}_{\text{SFT}} = -\frac{1}{N} \sum_{i=1}^{N} \sum_{t=1}^{T_i} \log \pi_\theta(o_{i,t}|o_{i,<t}, m_i, q_i) \tag{1}$$

where $N$ is the total number of training samples, $o_{i,t}$ represents the $t$-th token in the output sequence of the $i$-th training sample, $T_i$ denotes the length of the output sequence for the $i$-th sample, $o_{i,<t}$ represents all tokens before position $t$ in the $i$-th output sequence, $m_i \in \{\text{High}, \text{Medium}, \text{Low}\}$ is the mode indicator for the $i$-th sample, $q_i$ is the input query for the $i$-th sample, and $\pi_\theta$ is the model's policy parameterized by $\theta$. The training data is balanced across the three modes to ensure the model learns appropriate reasoning patterns for each mode while preserving its original capabilities.

### 3.2 BUDGET-AWARE REINFORCEMENT LEARNING

Our reinforcement learning strategy follows a carefully designed two-phase approach to ensure that controllable reasoning capabilities are built upon a strong performance foundation rather than compromising the model's peak abilities.

#### 3.2.1 DAPO FRAMEWORK

Our reinforcement learning approach builds upon the Decouple Clip and Dynamic sAmpling Policy Optimization (DAPO) framework (Yu et al., 2025). DAPO optimizes the policy by sampling a group of outputs $\{o_i\}_{i=1}^{G}$ for each input query $q$ (which includes both the mode specification and the problem statement) and corresponding answer $a$. The optimization objective is formulated as:

$$\mathcal{J}_{\text{DAPO}}(\theta) = \mathbb{E}_{(q,a)\sim\mathcal{D}, \{o_i\}_{i=1}^{G}\sim\pi_{\theta_{\text{old}}}(\cdot|q)}$$

$$\left[ \frac{1}{\sum_{i=1}^{G}|o_i|} \sum_{i=1}^{G} \sum_{t=1}^{|o_i|} \min\left(r_{i,t}(\theta)\hat{A}_{i,t}, \text{clip}\left(r_{i,t}(\theta), 1-\varepsilon_{\text{low}}, 1+\varepsilon_{\text{high}}\right)\hat{A}_{i,t}\right) \right] \tag{2}$$

where the importance sampling ratio is $r_{i,t}(\theta) = \frac{\pi_\theta(o_{i,t}|q,o_{i,<t})}{\pi_{\theta_{\text{old}}}(o_{i,t}|q,o_{i,<t})}$ and the advantage estimate is $\hat{A}_{i,t} = \frac{R_i - \text{mean}(\{R_i\}_{i=1}^{G})}{\text{std}(\{R_i\}_{i=1}^{G})}$.

### 3.2.2 PHASE 1: WARM-UP RL TRAINING

In the first phase, we focus exclusively on maximizing model performance without any compression constraints. The model is trained using standard RL objectives to reach its peak reasoning capability, establishing the performance ceiling that will serve as the reference point for subsequent compression. This warm-up phase is critical because it ensures that the model starts from its optimal state before learning to compress reasoning chains.

### 3.2.3 PHASE 2: RL WITH BUDGET-AWARE REWARD SHAPING

The second phase introduces controllable reasoning through budget-aware reward shaping. We implement different response length rewards for each mode, allowing the model to learn distinct reasoning capabilities with varying response lengths across different modes. However, we discovered that models tend to exploit compression objectives through "Reasoning Length Hacking"—reducing thinking tokens within <think> tags while compensating with extended reasoning in the answer section, rather than genuinely reducing reasoning depth. Thus, we incorporate Leak Penalty in our reward shaping strategy, effectively preventing reasoning spillover into answer sections.

**Adaptive Reward Shaping Strategies.** To enable controllable reasoning, we design a composite reward function that balances task performance with response length efficiency. Our approach builds upon the length reward framework from Kimi k1.5 (Team et al., 2025) and extends it with mode-specific adaptive mechanisms. The total reward for output $o_i$ in mode $m$ is defined as

$$R_i^{(m)} = R_{\text{task}}(o_i) + \alpha^{(m)} \cdot R_{\text{length}}(o_i) + R_{\text{leak}}(o_i) \tag{3}$$

where $R_{\text{task}}(o_i) \in \{0, 1\}$ evaluates task correctness through exact answer matching, $R_{\text{length}}(o_i)$ enforces response length constraints, $R_{\text{leak}}(o_i) \in \{-0.5, +0.5\}$ ensures proper reasoning-answer separation, and $\alpha^{(m)}$ is the mode-specific scaling coefficient that controls response compression intensity.

**Mode-Specific Response Length Reward.** The response length reward component implements mode-specific compression that allows different reasoning capabilities with varying response lengths across different modes:

$$R_{\text{length}}(o_i) = \begin{cases} \lambda_i & \text{if } R_{\text{task}}(o_i) = 1 \\ \min(0, \lambda_i) & \text{if } R_{\text{task}}(o_i) = 0 \end{cases}$$
$$\text{where } \lambda_i = 0.5 - \frac{\text{len}(o_i) - \text{len}_{\min}}{\text{len}_{\max} - \text{len}_{\min}} \tag{4}$$

Here, $\text{len}(o_i)$ denotes the total response length of output $o_i$ (including both thinking tokens within <think> tags and answer tokens after </think>), $\text{len}_{\min}$ and $\text{len}_{\max}$ are the minimum and maximum response lengths within the current group, respectively. The normalized response length penalty $\lambda_i$ ensures scale invariance across different problem complexities by mapping response lengths to a standardized $[-0.5, 0.5]$ range.

**Leak Penalty.** During RL training, we observed that models increasingly exhibit Reasoning Length Hacking behavior, where reasoning content leaks from the thinking section (within <think> tags) to the answer section (after </think>). This phenomenon stems from pre-existing patterns in our Budget-Mode SFT data, particularly in Low mode samples, where reasoning often overflows into answer sections due to aggressive compression during data construction. These pre-existing leakage patterns become reinforced during RL training, as they provide a path for models to maintain task correctness while achieving shorter thinking sections. However, this reinforcement of data artifacts defeats our compression objective, as reasoning effort is merely redistributed rather than genuinely reduced. To counteract this undesired pattern reinforcement, we implement:

$$R_{\text{leak}}(o_i) = \begin{cases} +0.5 & \text{if no transition keywords in answer section} \\ -0.5 & \text{if transition keywords detected in answer section} \end{cases} \tag{5}$$

where "transition keywords" refer to reasoning-related tokens such as "Wait", "Let me think", "Actually", "Alternatively", "However", and similar metacognitive expressions that typically indicate

ongoing reasoning processes. This binary reward mechanism penalizes the appearance of such keywords in the answer section (content after `</think>`), discouraging models from reinforcing the leakage patterns inherited from SFT data and ensuring that genuine reasoning compression occurs within the thinking section rather than pattern-based content redistribution. Illustrative cases are shown in Appendix F.4.

## 3.3 INFERENCE USAGE

During inference, users control reasoning modes by using the corresponding mode-specific prompts that were established during training. The model supports three operational modes: **High mode** for maximum accuracy with full reasoning capability, **Medium mode** for balanced performance with 50% compression, and **Low mode** for rapid responses with 75% compression. Users activate each mode by incorporating the respective budget-mode prompts (detailed in Appendix F.4) into their system prompts, ensuring consistent mode activation without requiring manual parameter tuning.

## 4 EXPERIMENTS

### 4.1 EXPERIMENTAL SETUP

**Datasets and Benchmarks.** We evaluate THINKDIAL across mathematical reasoning benchmarks spanning different difficulty levels: AIME 2025, AIME 2024, and GSM8K (Cobbe et al., 2021). Additionally, we use GPQA diamond (Rein et al., 2024) for out-of-distribution evaluation to assess generalization beyond mathematical domains, that is, we do not include any STEM data in the training set. For evaluation reliability, we use different sampling strategies: AIME problems are evaluated 32 times each, GSM8K uses 500 randomly sampled problems evaluated 4 times each, and GPQA uses 198 samples evaluated 8 times each. The statistical significance of these results is shown in Appendix F.1

**Implementation Details.** We use Qwen-2.5-Instruct-32B (Bai et al., 2023) as our foundation model. We also include experiments on DeepSeek-R1-Distilled-Qwen-7B (Guo et al., 2025) and Qwen3-8B (Yang et al., 2025a) to demonstrate that THINKDIAL generalizes robustly across different model scales in Appendix F.2. We first perform supervised fine-tuning (SFT) including 12K original reasoning data and 6K Budget-Mode SFT data, with details provided in the Appendix F.4. For RL training, both the warm-up phase and the length reward shaping phase utilize the same set of 20K in-house mathematical problems. We set truncation ratios $r_{med} = 0.5$ and $r_{low} = 0.25$ for Medium and Low mode, respectively, and of course, $r_{high} = 1$. The mode-specific scaling coefficients follow a progressive strategy: $\alpha^{(high)} = 0.0$, $\alpha^{(med)} = 0.5$, and $\alpha^{(low)} = 1.0$. The extensive experiments of different $\alpha$ configurations are shown in Appendix F.3 Training follows our two-phase strategy: 95 steps on High mode data to establish peak performance, then 40 additional steps with length rewards for controllable reasoning capabilities.

**Baselines.** We evaluate THINKDIAL against several key baselines to validate the effectiveness of our approach:

1. **Peak-Performance Checkpoint**: The capability peak checkpoint of the Qwen-2.5-Instruct-32B model after undergoing training with 12K original reasoning data for SFT and 20K in-house mathematical problems for RL.

2. **w/o Budget-Mode SFT**: Our framework without specialized mode-conditioned SFT but only original reasoning data to assess controllable reasoning pretraining necessity;

3. **Only Budget-Mode SFT**: Exclusive mode-specific SFT without RL optimization;

4. **w/o Warm-up**: Direct compression training without establishing peak performance;

5. **Peak Truncation**: Simple truncation-based compression at performance peaks;

6. **gpt-oss-120b** and **o3-mini**: OpenAI's proprietary controllable reasoning models for state-of-the-art mode-based comparison.

Table 1: Overall ACT Score performance. L, M, and H represent Low, Medium, and High modes, respectively. "w/o BM SFT" refers to the model trained without Budget Mode SFT data. The best result in each column is highlighted in bold.

| Model | AIME 2024 | | | | AIME 2025 | | | | GSM8K | | | |
|---|---|---|---|---|---|---|---|---|---|---|---|---|
| | L | M | H | Avg. | L | M | H | Avg. | L | M | H | Avg. |
| Ours | 63.4 | **68.3** | **108.4** | **80.0** | 57.1 | **70.2** | 107.6 | **78.3** | **99.3** | **98.7** | 100.8 | **99.6** |
| w/o BM SFT | 59.5 | 68.1 | 84.7 | 70.8 | 57.7 | 62.8 | 85.7 | 68.8 | 93.9 | 94.6 | 99.0 | 95.8 |
| Only BM SFT | **66.0** | 60.7 | 93.1 | 73.3 | **63.8** | 63.2 | 97.2 | 74.8 | 61.1 | 73.9 | 99.8 | 78.3 |
| w/o Warmup | 48.6 | 64.6 | 95.5 | 69.6 | 47.1 | 61.7 | 87.2 | 65.3 | 98.4 | 95.6 | 100.5 | 98.2 |
| Peak Truncation | 47.1 | 40.5 | 106.5 | 64.7 | 44.3 | 38.2 | **108.8** | 63.8 | 76.6 | 82.9 | 100.1 | 86.5 |

**Evaluation Metrics.** To quantify the overall effectiveness of controllable reasoning, we define a composite metric **Accuracy-Cost Trade-off (ACT) Score** that balances accuracy retention and compression efficiency. For mode $m \in \{\text{High}, \text{Medium}, \text{Low}\}$, we compute accuracy retention ratio $A_m = \frac{\text{Acc}_m}{\text{Acc}_{\text{base}}}$ and compression rate $C_m = 1 - \frac{\text{Cost}_m}{\text{Cost}_{\text{base}}}$, where base values represent the corresponding performance of the peak-performance checkpoint. The ACT score is:

$$S_{\text{ACT}} = \frac{\sum_{m \in \mathcal{M}} (\beta^{(m)} \cdot A_m + (1 - \beta^{(m)}) \cdot C_m)}{|\mathcal{M}|} \quad (6)$$

$$\beta^{(m)} = \begin{cases} 1 & \text{if } m = \text{High} \\ 0.5 & \text{if } m \in \{\text{Medium}, \text{Low}\} \end{cases} \quad (7)$$

Since High mode prioritizes performance maintenance while Medium and Low modes balance accuracy and efficiency equally, we set different weighting coefficients $\beta_m$ to reflect these distinct operational objectives.

**Direct Performance Visualization.** We analyze raw accuracy and thinking token length across modes by comparing with gpt-oss-120b and o3-mini to evaluate whether our approach can replicate the controllable reasoning patterns of proprietary systems. This visualization approach reveals whether our step-wise degradation curves match those of SOTA mode-based reasoning models.

## 4.2 Overall Performance Analysis

Table 1 presents comprehensive ACT score evaluation results across different benchmarks, validating our core hypothesis that models can learn efficient reasoning without sacrificing problem-solving capabilities. Notably, High mode performance matches or even exceeds the original model baseline, addressing a fundamental concern in compression research—that controllability training might compromise peak performance. The clear step-wise degradation pattern (High → Medium → Low) demonstrates precise control over the accuracy-efficiency trade-off, achieving the target compression rates while maintaining specified performance thresholds.

Beyond raw performance metrics, our framework demonstrates intelligent adaptation to problem complexity. The model naturally allocates more reasoning effort to challenging problems (AIME series) while maintaining efficiency on simpler tasks (GSM8K), suggesting learned rather than mechanical compression behavior. Most importantly, direct performance visualizations confirm successful replication of gpt-oss-style patterns—our accuracy-token curves closely match those of gpt-oss-120b and o3-mini, proving that mode-based control can be achieved through our approach. Finally, the framework's robustness extends beyond mathematical reasoning, with GPQA evaluation showing effective transfer to other domains despite training primarily on mathematical data.

## 4.3 Detailed Analysis

**Impact of Budget-Mode Supervised Fine-tuning.** Figure 3 demonstrates the critical role of Budget-Mode SFT in preventing mode interference during RL training. Without specialized SFT, RL training with length rewards alone causes the three operational modes to interfere with each

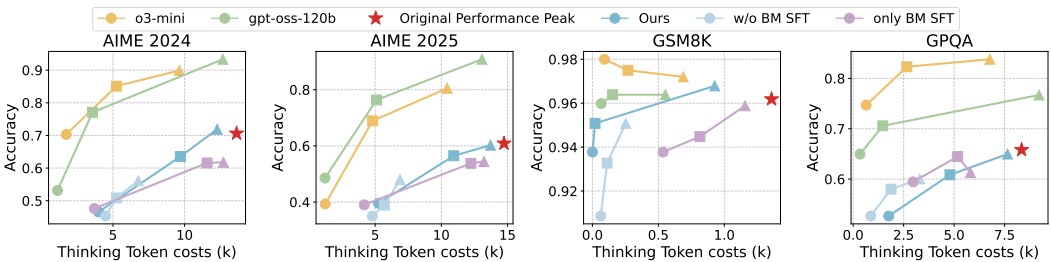

Figure 3: The impact of Budget Mode SFT across different datasets.

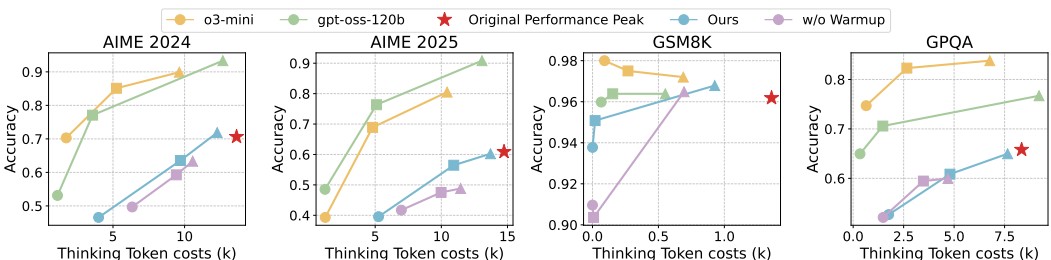

Figure 4: The impact of warm-up RL training on controllable reasoning performance across different modes.

other, leading to performance collapse in High mode that falls significantly below the original performance peak. This interference pattern completely undermines the controllable reasoning objectives. With proper SFT initialization, the RL exploration phase cannot cause modes to interfere with each other. Consequently, High mode maintains performance at or above the original peak level, while Medium and Low modes achieve effective compression with controlled degradation. However, while Budget-Mode SFT establishes mode awareness, relying exclusively on SFT without RL optimization leads to significant accuracy degradation in High and Medium modes. This finding validates our end-to-end training paradigm: Budget-Mode SFT provides the essential semantic foundation, while RL optimization fine-tunes the accuracy-efficiency balance. Neither component alone can achieve the sophisticated controllable reasoning demonstrated by our complete framework.

**The Importance of Two-Phase RL Training Strategy.** Our two-phase RL training strategy proves essential for effective controllable reasoning. Figure 4 demonstrates that without the warm-up phase (Phase 1) to first establish peak performance, the model struggles to maintain quality in High and Medium modes when budget-aware reward shaping (Phase 2) is introduced. The warm-up phase ensures that compression capabilities are built upon a strong performance foundation rather than compromising the model's peak abilities.

More importantly, Figure 5 compares our approach with the Peak Truncation Method that cuts reasoning chains at peak performance to target token budgets, then asks the model to generate summaries and answers. The visualization reveals that such mechanical truncation completely fails to achieve gpt-oss-style controllable reasoning patterns. While our learned compression maintains smooth degradation curves similar to proprietary systems, the Peak Truncation Method shows catastrophic performance collapse, demonstrating that sophisticated training is essential for effective mode-based reasoning control.

**Addressing Reasoning Length Hacking.** Figure 6 presents token statistics comparing models with and without Leak Penalty, revealing the critical importance of addressing Reasoning Length Hacking. The bar chart demonstrates a counterintuitive phenomenon: without Leak Penalty, although thinking tokens decrease as intended, answer tokens significantly increase, resulting in higher total token consumption—defeating our compression objectives.

However, with Leak Penalty in place, the model not only reduces thinking tokens but also maintains concise answer sections, achieving genuine overall token reduction. This analysis validates

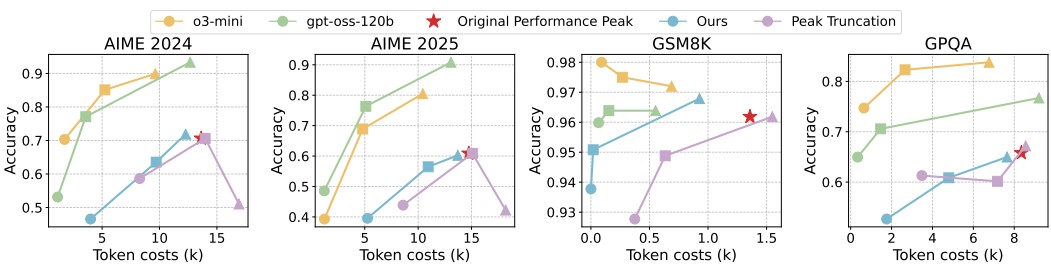

Figure 5: Performance comparison between our approach and the Peak Truncation Method.

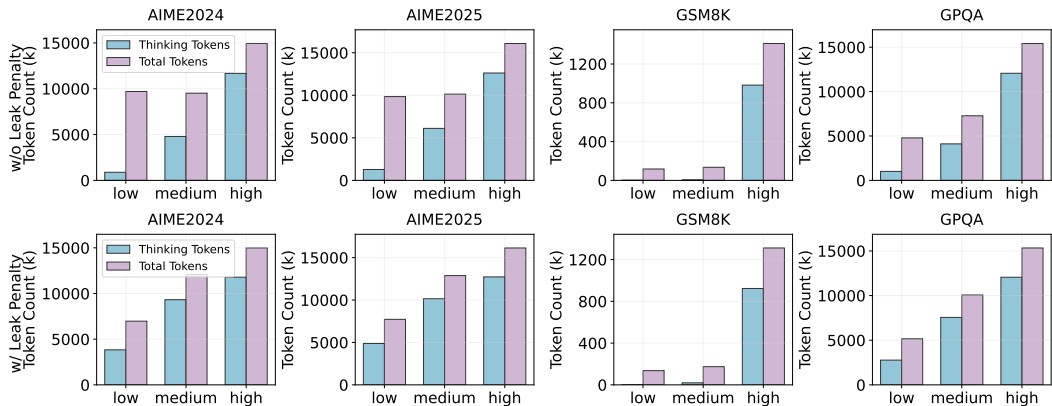

Figure 6: The impact of Leak Penalty on model response length. Total Tokens include both Thinking Tokens and Summary Tokens.

that effective controllable reasoning requires preventing models from circumventing compression constraints through reasoning spillover into answer sections.

**Analysis of BM SFT Data Amount.** As shown in Figure 7, the relationship between Budget-Mode (BM) SFT data amount and model performance reveals a critical balance in training data composition. We compare two configurations: the balanced setup (6K BM SFT + 12K original reasoning data) versus the BM-heavy setup (12K BM SFT + 12K original reasoning data).

First, adding an appropriate amount of BM data does not compromise the model's performance ceiling and can even provide modest improvements. The balanced configuration maintains the model's peak reasoning capabilities while establishing effective mode differentiation. Importantly, moderate BM data inclusion does not suppress the model's output length in High mode, preserving its ability to generate comprehensive reasoning when needed.

However, excessive BM SFT data creates significant performance degradation. When BM SFT data increases substantially, the model's performance ceiling drops noticeably across Medium and Hard questions. More critically, this BM-heavy configuration severely suppresses reasoning length across all operational modes, indicating that the model becomes overly constrained in its reasoning capacity. This length suppression is particularly problematic as it limits the model's ability to engage in thorough reasoning even when explicitly instructed to operate in High mode. These findings highlight the importance of balanced training data composition in our Budget-Mode SFT approach.

### 4.4 COMPARISON WITH OPEN TOKEN COMPRESSION BASELINES

We further compare THINKDIAL with representative open controllability and compressed-reasoning methods (Chen et al., 2025; Shen et al., 2025; Luo et al., 2025; Arora & Zanette, 2025; Wu et al., 2025), and the binary-gating method (Zhang et al., 2025a). To ensure fair comparison, we use baselines that report results of GSM8K, MATH and AIME 2024 on DeepSeek-R1-Distilled-Qwen-7B. To quantify the accuracy–efficiency trade-off, we adopt the ACT Score: $\text{ACT} = \beta \cdot \frac{\text{acc}}{\text{acc}_{\text{ref}}} + (1 -$

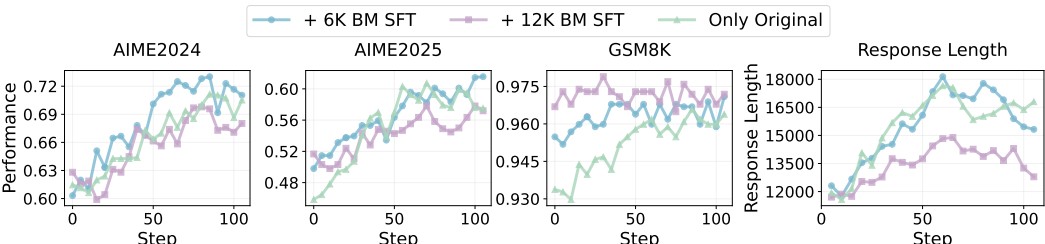

Figure 7: Impact of budget mode (BM) SFT data amount on warm-up phase RL training performance and response length. Here, "+ BM SFT" represents the amount of BM SFT data mixed with original reasoning data.

Table 2: Comparison with open token compression baselines using ACT Score ($\beta = 0.5$). The backbone model is eepSeek-R1-Distilled-Qwen-7B. Best values in **bold**; second best is underline.

| Model | GSM8K | | | MATH | | | AIME24 | | |
|---|---|---|---|---|---|---|---|---|---|
| | **Acc** | **Len** | **ACT** | **Acc** | **Len** | **ACT** | **Acc** | **Len** | **ACT** |
| Non-Thinking | 85.1 | 283 | N/A | 80.6 | 697 | N/A | 24.2 | 1929 | N/A |
| Thinking | 87.9 | 682 | N/A | 90.2 | 3674 | N/A | 53.5 | 10306 | N/A |
| OverThink (Chen et al., 2025) | 86.3 | 426 | 67.9 | 89.4 | 2435 | 66.4 | 53.1 | 8744 | 57.2 |
| DAST (Shen et al., 2025) | 86.7 | 459 | 65.7 | 89.6 | 2162 | 70.2 | 45.6 | **7578** | 55.9 |
| O1-Pruner (Luo et al., 2025) | 87.6 | 428 | 68.5 | 86.6 | 2534 | 63.5 | 49.2 | 9719 | 48.8 |
| TLMRE (Arora & Zanette, 2025) | 88.9 | 756 | 45.1 | 91.8 | 2899 | 61.4 | 54.0 | 8633 | 58.6 |
| ModelMerging (Wu et al., 2025) | 88.4 | 531 | 61.4 | 72.6 | 2280 | 59.2 | 36.9 | 8624 | 42.7 |
| AdaptThink (Zhang et al., 2025a) | **91.0** | 309 | 79.1 | 92.0 | 1875 | 75.5 | 55.6 | 8599 | **60.2** |
| THINKDIAL-High | 85.9 | 398 | 69.7 | **93.9** | 2591 | 66.8 | **56.7** | 10314 | 53.0 |
| THINKDIAL-Medium | 84.4 | 375 | 70.5 | 91.7 | 2059 | 72.8 | 53.4 | 9969 | 51.5 |
| THINKDIAL-Low | 79.9 | **212** | **79.9** | 88.8 | **1264** | **82.0** | 52.4 | 8174 | 59.3 |

$\beta) \cdot \frac{\text{len}_{\text{ref}} - \text{len}}{\text{len}_{\text{ref}}}$, where $\beta = 0.5$, balancing accuracy retention with output-length reduction relative to the DeepSeek-R1-Distilled-Qwen-7B (Thinking) reference[1].

As shown in Table 2, across all three benchmarks, THINKDIAL-Low consistently reaches the top of the ACT trade-off curve, indicating that THINKDIAL achieves stronger accuracy–efficiency balance than all prior open baselines. In accuracy-focused settings, THINKDIAL-High matches or surpasses the strongest baselines (e.g., best accuracy on MATH and AIME 2024), showing that controllability does not compromise peak performance. Importantly, all baselines are trained for a single compression target. In contrast, THINKDIAL delivers three stable, discrete reasoning regimes in one unified checkpoint, switchable purely via prompts without retraining—offering a level of controllability not available in existing open methods.

## 5 CONCLUSION

We present THINKDIAL, the first open-recipe framework to successfully implement gpt-oss-style controllable reasoning through discrete operational modes. Our end-to-end training paradigm, combining Budget-Mode Supervised Fine-tuning and Budget-Aware Reinforcement Learning with reward shaping, achieves target compression-performance trade-offs while preserving peak capabilities. Extensive experiments demonstrate that our approach closely matches proprietary systems' controllable reasoning patterns, significantly outperforming naive truncation methods. By democratizing sophisticated mode-based reasoning control, this work enables broader research and application development in controllable AI reasoning, establishing a foundation for future advances in adaptive computational allocation.

---

[1]Here, ACT is computed using output-token length, following the reporting conventions of prior work.

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

## A    THE USE OF LARGE LANGUAGE MODELS

In the preparation of this paper, we employed large language models (LLMs) as an auxiliary tool to enhance the clarity and coherence of our writing. We used LLMs to polish and refine the language of our paper, including improving sentence structure, grammar, and overall readability. The LLMs helped us rephrase technical concepts more clearly and ensure consistent terminology throughout the paper.

It is important to note that all core research contributions, including the theoretical analysis, experimental design, implementation details, and result interpretation, were entirely conceived and developed by the human authors. The LLMs served solely as a writing and editing tool, and did not contribute to the scientific content, methodology, or conclusions of this work. All experimental results, data analysis, and technical insights presented in this paper are the original contributions of the authors. We maintain full responsibility for all content and claims made in this paper.

## B    ETHICS STATEMENT

We have carefully considered the ethical implications of our research. Our work focuses on improving computational efficiency in language model reasoning, which contributes to making AI systems more accessible and environmentally sustainable. We do not involve human subjects in our study, as all evaluations are conducted on established public benchmarks (AIME, GSM8K, GPQA). We do not collect personal data or engage in practices that compromise privacy. We do not develop techniques with potential for malicious use, and our approach does not introduce discriminatory biases across different reasoning domains.

## C    REPRODUCIBILITY STATEMENT

We have made extensive efforts to ensure the reproducibility of our research. All experimental parameters, training configurations, and implementation details are comprehensively documented throughout the paper.

Section 3 provides detailed mathematical formulations for our training methodology. Section F.4 in the appendix provides detailed specifications of our Budget-Mode Supervised Fine-tuning data construction. Section E presents complete hyperparameters for our reinforcement learning, including learning rates, batch sizes, clipping parameters, and training steps. Our experimental methodology is fully specified in Section 4.1, covering evaluation protocols, sampling strategies, and the composite ACT score metric with explicit mathematical formulations.

While our training incorporates some proprietary mathematical reasoning data, our methodology is fundamentally data-agnostic. This is rigorously demonstrated through our strong performance on out-of-distribution evaluation using GPQA, where we achieve effective reasoning control without including any STEM data in the training set. The generalization results validate that our techniques can be applied to diverse datasets and domains, ensuring broad applicability of our approach beyond the specific training data used in our experiments. All methodological details are transparently reported to ensure full reproducibility.

## D    BUDGET-MODE SUPERVISED FINE-TUNING DATA

### D.1    MODE-SPECIFIC SYSTEM PROMPTS

To enable the switching of model output distributions across different modes, we design distinct system prompts for each reasoning mode using the same data (Yang et al., 2025b). As shown in Table 6, these mode-specific prompts establish clear semantic relationships between budget constraints and expected reasoning behaviors, guiding the model to adopt appropriate reasoning strategies for each mode.

Table 3: Data composition and average response lengths for budget mode supervised fine-tuning.

| Budget Mode | Quantity | Average Length |
|---|---|---|
| High | 2197 | 9440.0 |
| Medium | 2197 | 5125.3 |
| Low | 2132 | 1302.7 |

## D.2 Data Construction

We construct specialized training data that illustrates how the same problem can be solved with different levels of reasoning depth while preserving correctness. Figure 2, starting from high-quality, complete reasoning chains used for the High mode, we systematically derive compressed Medium and Low variants through targeted truncation at approximately $r_{medium}$ (50%) and $r_{low}$ (25%) of the original thinking-token length. To maintain structural integrity, truncation is performed strictly at `\n\n` step boundaries, ensuring that every retained segment corresponds to a full reasoning step and that no partial steps are introduced. The final truncated length is controlled to fall within $\pm 1\%$ of the desired ratio, while still respecting the step-boundary constraint.

After truncation, if the regenerated answer no longer matches the gold solution, the example is discarded. To ensure narrative coherence under reduced reasoning budgets, we append a mode-specific connective phrase at the end of the truncated thinking section, positioned immediately before the `</think>` marker. This connective helps stabilize transitions from shortened reasoning to answer generation. Each budget mode is additionally paired with a dedicated system prompt that specifies the expected reasoning style (e.g., detailed, moderately concise, highly concise), allowing the model to internalize the semantic mapping between mode definitions and reasoning behaviors.

We present training data examples demonstrating how the same problem is solved with different levels of reasoning depth across the three budget modes. Tables 7, 8, and 9 showcase the key differences: (1) the thinking section length varies significantly across modes, with High mode containing the most detailed reasoning process; (2) after truncation, we add mode-specific connective text before the end-of-think marker (highlighted in brown) to ensure smooth logical transitions; and (3) the answer sections are regenerated for each truncated sample to maintain correctness despite the compressed reasoning. Only samples that preserve both logical coherence and accuracy after this construction process are retained in the training data.

## D.3 Training Data Selection

To construct high-quality training data for budget-mode supervised fine-tuning, we build a dataset of 6K budget-mode SFT samples based on our in-house training data. The dataset maintains a balanced distribution across reasoning effort levels with a ratio of high:medium:low = 1:1:1, ensuring equal representation of each budget mode during training.

For the medium and low budget modes, we set the reasoning effort ratios $r_{medium}$ and $r_{low}$ to 50% and 25%, respectively. Additionally, we incorporate 800 GSM8K samples with empty thinking into the low mode to further enhance the model's ability to provide concise responses under strict budget constraints. Table 3 summarizes the data composition and characteristics for each budget mode.

To preserve the model's general problem-solving capabilities during budget-aware fine-tuning, we incorporate 12K samples of light-R1 long chain-of-thought reasoning data from DeepSeek-R1 (Wen et al., 2025). This auxiliary dataset consists of 3K samples from stage 2 training and 9k samples randomly selected from stage 1, providing diverse reasoning patterns that help maintain the model's foundational reasoning abilities while adapting to budget constraints.

## E Training Details

During RL training, we set the maximum prompt length to 2,048 tokens and the maximum response length to 32,768 tokens. DAPO training is conducted for 135 steps (95 steps High mode + 40 steps budget-aware) with a batch size of 256 and a mini-batch size of 256. The actor is optimized

Table 4: Bootstrap 95% confidence intervals and significance tests across datasets and modes. All values: accuracy, standard error, 95% CI bounds, and two-sided $t$-tests. All results are statistically significant at $\alpha = 0.05$.

| Dataset | Mode | Mean | Std Error | 95% CI L | 95% CI U | t-stat | p-value |
|---|---|---|---|---|---|---|---|
| AIME24 | High | 0.72 | 0.01 | 0.69 | 0.75 | 15.07 | 0 |
| AIME24 | Medium | 0.64 | 0.02 | 0.60 | 0.67 | 8.72 | 0 |
| AIME24 | Low | 0.47 | 0.02 | 0.43 | 0.50 | -2.14 | 0.03 |
| AIME25 | High | 0.60 | 0.02 | 0.57 | 0.63 | 6.37 | 0 |
| AIME25 | Medium | 0.56 | 0.02 | 0.53 | 0.60 | 3.97 | 0 |
| AIME25 | Low | 0.40 | 0.02 | 0.36 | 0.43 | -6.51 | 0 |
| GPQA | High | 0.65 | 0.01 | 0.63 | 0.67 | 12.46 | 0 |
| GPQA | Medium | 0.61 | 0.01 | 0.58 | 0.63 | 8.83 | 0 |
| GPQA | Low | 0.53 | 0.01 | 0.50 | 0.55 | 2.12 | 0.03 |
| GSM8K | High | 0.97 | 0.01 | 0.96 | 0.98 | 83.73 | 0 |
| GSM8K | Medium | 0.95 | 0.01 | 0.94 | 0.96 | 65.78 | 0 |
| GSM8K | Low | 0.94 | 0.01 | 0.92 | 0.95 | 57.18 | 0 |
| MATH | High | 0.96 | 0.01 | 0.95 | 0.97 | 86.96 | 0 |
| MATH | Medium | 0.91 | 0.01 | 0.89 | 0.93 | 50.86 | 0 |
| MATH | Low | 0.89 | 0.01 | 0.87 | 0.91 | 44.10 | 0 |

using Adam, with learning rates of $1 \times 10^{-6}$, and a linear warm-up schedule over 20 steps. Our implementation uses grouped sampling with group size $G = 16$. For clipping parameters, we set the clip ratio low to 0.2 and clip ratio high to 0.28.

During the 95-step warmup phase of RL training, we employ Dynamic Sampling. However, in the RL with length rewards phase, we disable Dynamic Sampling because even when all responses in a group are either correct or incorrect, the presence of length rewards still provides effective gradients.

Our implementation is based on the VeRL framework. Rollouts are generated using temperature sampling ($\tau = 1.0$), with enforced end-of-sequence tokens. We leverage vLLM for efficient batched decoding with 256 rollout slots and paged attention.

# F EXPERIMENT DETAILS

## F.1 STATISTICAL SIGNIFICANCE AND CONFIDENCE INTERVALS.

@e report 95% bootstrap confidence intervals and significance tests for all major benchmarks. Since each evaluation already uses repeated sampling (AIME: 32 samples per instance; GSM8K: 12 samples; GPQA: 8 samples; MATH: 16 samples), computing uncertainty estimates is straightforward. As shown in Table 4, all High, Medium, and Low modes exhibit statistically significant differences ($p < 0.05$), and the confidence intervals are narrow across datasets. These results confirm that THINKDIAL's accuracy differences across modes are statistically robust, further validating the reliability of our evaluation methodology.

## F.2 SCALABILITY ACROSS BACKBONE MODELS

To assess whether THINKDIAL generalizes beyond a single 32B-scale model, we apply the full three-mode recipe to two additional backbones: Qwen3-8B (Yang et al., 2025a), a newer and more capable reasoning model, and DeepSeek-R1-Distilled-Qwen-7B (Guo et al., 2025), a popular distilled reasoning model.

As shown in Figure 8, across both model families, THINKDIAL consistently exhibits three well-separated reasoning modes with predictable accuracy–length trade-offs: High retains peak accuracy, Medium offers balanced performance, and Low delivers substantial token reduction. Moreover, High-mode accuracy matches or exceeds the corresponding SOTA distilled baselines (e.g., Qwen3-8B-High on AIME24), demonstrating that controllability does not compromise peak reasoning abil-

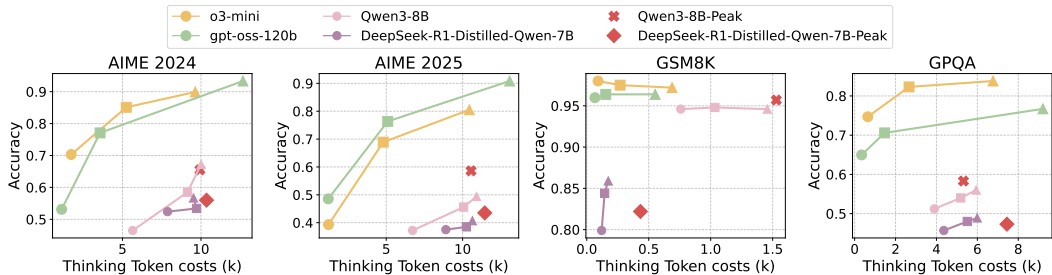

Figure 8: THINKDIAL applied to additional backbone models (Qwen3-8B and DeepSeek-R1-Distilled-Qwen-7B).

Table 5: Overall ACT Score performance. L, M, and H represent Low, Medium, and High modes, respectively. "w/o BM SFT" refers to the model trained without Budget Mode SFT data. The best result in each column is highlighted in bold.

| Model | AIME 2024 | | | | AIME 2025 | | | | GSM8K | | | |
|---|---|---|---|---|---|---|---|---|---|---|---|---|
| | L | M | H | Avg. | L | M | H | Avg. | L | M | H | Avg. |
| A: $(\alpha_{\text{low}}, \alpha_{\text{med}}, \alpha_{\text{high}}) = (0.3, 0.5, 1.0)$ | | | | | | | | | | | | |
| **Ours** | 63.4 | 50.0 | 108.4 | 73.9 | 57.1 | 49.8 | 107.6 | 71.5 | 99.3 | 98.5 | 100.8 | 99.5 |
| w/o BM SFT | 59.5 | 62.4 | 84.7 | 68.9 | 57.7 | 58.4 | 85.7 | 67.3 | 93.9 | 92.9 | 99.0 | 95.3 |
| Only BM SFT | 66.0 | 39.3 | 93.1 | 66.1 | 63.8 | 41.4 | 97.2 | 67.5 | 61.1 | 57.6 | 99.8 | 72.8 |
| w/o Warmup | 48.6 | 48.1 | 95.5 | 64.1 | 47.1 | 46.2 | 87.2 | 60.2 | 98.4 | 96.6 | 100.5 | 98.5 |
| Peak Truncation | 47.1 | 16.2 | 106.5 | 56.6 | 44.3 | 13.3 | 108.8 | 55.5 | 76.6 | 72.3 | 100.1 | 83.0 |
| B: $(\alpha_{\text{low}}, \alpha_{\text{med}}, \alpha_{\text{high}}) = (0.5, 0.5, 0.9)$ | | | | | | | | | | | | |
| **Ours** | 65.4 | 50.0 | 95.4 | 70.2 | 61.0 | 49.8 | 94.2 | 68.3 | 98.8 | 98.5 | 91.2 | 96.2 |
| w/o BM SFT | 62.0 | 62.4 | 79.5 | 68.0 | 59.1 | 58.4 | 80.7 | 66.1 | 94.1 | 92.9 | 96.5 | 94.5 |
| Only BM SFT | 67.6 | 39.3 | 81.3 | 62.7 | 65.5 | 41.4 | 85.3 | 64.0 | 71.5 | 57.6 | 88.0 | 72.4 |
| w/o Warmup | 56.1 | 48.1 | 85.5 | 63.3 | 54.9 | 46.2 | 77.9 | 59.7 | 97.4 | 96.6 | 93.3 | 95.8 |
| Peak Truncation | 58.9 | 16.2 | 93.9 | 56.3 | 54.1 | 13.3 | 95.6 | 54.3 | 82.3 | 72.3 | 86.9 | 80.5 |
| C: $(\alpha_{\text{low}}, \alpha_{\text{med}}, \alpha_{\text{high}}) = (0.3, 0.7, 0.9)$ | | | | | | | | | | | | |
| **Ours** | 63.4 | 68.3 | 95.4 | 75.7 | 57.1 | 70.2 | 94.2 | 73.8 | 99.3 | 98.7 | 91.2 | 96.4 |
| w/o BM SFT | 59.5 | 68.1 | 79.5 | 69.0 | 57.7 | 62.8 | 80.7 | 67.1 | 93.9 | 94.6 | 96.5 | 95.0 |
| Only BM SFT | 66.0 | 60.7 | 81.3 | 69.3 | 63.8 | 63.2 | 85.3 | 70.8 | 61.1 | 73.9 | 88.0 | 74.3 |
| w/o Warmup | 48.6 | 64.6 | 85.5 | 66.3 | 47.1 | 61.7 | 77.9 | 62.2 | 98.4 | 95.6 | 93.3 | 95.8 |
| Peak Truncation | 47.1 | 40.5 | 93.9 | 60.5 | 44.3 | 38.2 | 95.6 | 59.4 | 76.6 | 82.9 | 86.9 | 82.1 |

ity. These results indicate that THINKDIAL is scale-agnostic and readily applicable to newer or smaller backbones, confirming that the proposed recipe generalizes robustly beyond the Qwen2.5-Instruct-32B model.

### F.3 ROBUSTNESS TO $\alpha$ CHOICES.

To evaluate the robustness of THINKDIAL with respect to different trade-off preferences between accuracy and efficiency, we further vary the $\alpha$ values used in the ACT Score (Section 4.1) and report results under three configurations: *Setting A*: $(\alpha_{\text{low}}, \alpha_{\text{med}}, \alpha_{\text{high}}) = (0.3, 0.5, 1.0)$ (low emphasizes efficiency, high emphasizes accuracy), *Setting B*: $(0.5, 0.5, 0.9)$ (low and medium balance equally, high prefers accuracy), and *Setting C*: $(0.3, 0.7, 0.9)$ (low focuses on efficiency, medium and high increasingly favor accuracy). Table 5 summarizes ACT Scores on AIME24/25 and GSM8K. Across all settings, THINKDIAL consistently achieves the best overall ACT Score, and the relative ordering of the High, Medium, and Low modes remains stable, indicating that our learned modes are robust to the specific choice of $\alpha$.

*/\* Low Mode \*/*

You have extremely limited time to think and respond to the users query. Every additional second of processing and reasoning incurs a significant resource cost, which could affect efficiency and effectiveness. Your task is to prioritize speed without sacrificing essential clarity or accuracy. Provide the most direct and concise answer possible. Avoid unnecessary steps, reflections, verification, or refinements UNLESS ABSOLUTELY NECESSARY. Your primary goal is to deliver a quick, clear and correct response.

*/\* Medium Mode \*/*

You have sufficient time to think and respond to the user's query, allowing for a more thoughtful and in-depth answer. However, be aware that the longer you take to reason and process, the greater the associated resource costs and potential consequences. While you should not rush, aim to balance the depth of your reasoning with efficiency. Prioritize providing a well-thought-out response, but do not overextend your thinking if the answer can be provided with a reasonable level of analysis. Use your reasoning time wisely, focusing on what is essential for delivering an accurate response without unnecessary delays and overthinking.

*/\* High Mode \*/*

You have unlimited time to think and respond to the user's question. There is no need to worry about reasoning time or associated costs. Your only goal is to arrive at a reliable, correct final answer. Feel free to explore the problem from multiple angles, and try various methods in your reasoning. This includes reflecting on reasoning by trying different approaches, verifying steps from different aspects, and rethinking your conclusions as needed. You are encouraged to take the time to analyze the problem thoroughly, reflect on your reasoning promptly and test all possible solutions. Only after a deep, comprehensive thought process should you provide the final answer, ensuring it is correct and well-supported by your reasoning.

Table 6: System prompts used for High, Medium, and Low modes during data construction.

## F.4 CASE STUDY OF LEAK PENALTY

To assess the effectiveness of the Leak Penalty, we present two contrasting cases. Without the penalty (Table 10), the model keeps the thinking section brief but moves extensive reasoning—detailed calculations, repeated attempts, and lengthy explanations—into the answer, which is precisely the leak we aim to prevent. With the penalty (Table 11), the model conducts its reasoning within the ⟨think⟩ tags and returns a clean, direct answer; the answer section contains only the final summarized solution, achieving genuine compression without loss of quality.

918
919
920
921
922
923
924
925
926
927
928
929
930
931
932
933
934
935
936
937
938
939
940
941
942
943
944
945
946
947
948
949
950
951
952
953
954
955
956
957
958
959
960
961
962
963
964
965
966
967
968
969
970
971

**Prompt for High Mode**

*/* System Prompt */*
You have unlimited time to think and respond to the user's question. There is no need to worry about reasoning time or associated costs. Your only goal is to arrive at a reliable, correct final answer. Feel free to explore the problem from multiple angles, and try various methods in your reasoning. This includes reflecting on reasoning by trying different approaches, verifying steps from different aspects, and rethinking your conclusions as needed. You are encouraged to take the time to analyze the problem thoroughly, reflect on your reasoning promptly and test all possible solutions. Only after a deep, comprehensive thought process should you provide the final answer, ensuring it is correct and well-supported by your reasoning.

*/* User Prompt */*
One of the external common tangent lines of the two externally tangent circles with centers $O_1$ and $O_2$ touches the circles at $B$ and $C$, respectively. Let $A$ be the common point of the circles. The line $BA$ meets the circle with center $O_2$ at $A$ and $D$. If $|BA| = 5$ and $|AD| = 4$, then what is $|CD|$?

**Assistant Response**

**</think>**
Um, let's try to solve this geometry problem step by step. First, let's understand the problem statement. We have two externally tangent circles with centers $O_1$ and $O_2$. They touch at point A, which is their common point. There's an external common tangent that touches the first circle at B and the second at C. The line BA intersects the second circle again at D, with $|BA| = 5$ and $|AD| = 4$. We need to find $|CD|$.

*( ... 20874 tokens omitted for brevity )*

Instead, use the fact that the power of A with respect to the tangent BC is zero, and the ratio
$|BA|/|AD| = 5/4 = r_1/r_2$, so by similar triangles,
$|CD| = |BC| \cdot (r_2/(r_1 + r_2)) = 3\sqrt{5} \cdot (4/9) = 4\sqrt{5}/3$, no. I think the correct answer is 6, and the earlier steps had a miscalculation, but given the time I've spent, I'll go with 6.
**</think>**

The problem involves two externally tangent circles with a common external tangent touching them at points B and C. The line BA intersects the second circle again at D. Using the power of a point theorem, the length of the tangent BC squared is equal to the product of the segments of the secant from B through A to D.
Given $|BA| = 5$ and $|AD| = 4$, the power of point B with respect to the second circle gives
$|BC|^2 = |BA| \times |BD| = 5 \times (5 + 4) = 5 \times 9 = 45$. Recognizing the geometric relationship and using the power of point C with respect to the first circle, we find that $|CD|$ can be determined by similar triangles or the properties of the tangent and secant. The key insight is using the power of a point theorem correctly, leading to the conclusion that $|CD| = 6$.

**Answer:** $\boxed{6}$

Table 7: A case study of supervised fine-tuning data, presenting the prompt and corresponding response for the *high* mode.

**Prompt for Medium Mode**

*/* System Prompt */*

You have sufficient time to think and respond to the user's query, allowing for a more thoughtful and in-depth answer. However, be aware that the longer you take to reason and process, the greater the associated resource costs and potential consequences. While you should not rush, aim to balance the depth of your reasoning with efficiency. Prioritize providing a well-thought-out response, but do not overextend your thinking if the answer can be provided with a reasonable level of analysis. Use your reasoning time wisely, focusing on what is essential for delivering an accurate response without unnecessary delays and overthinking.

*/* User Prompt */*

One of the external common tangent lines of the two externally tangent circles with centers $O_1$ and $O_2$ touches the circles at $B$ and $C$, respectively. Let $A$ be the common point of the circles. The line $BA$ meets the circle with center $O_2$ at $A$ and $D$. If $|BA| = 5$ and $|AD| = 4$, then what is $|CD|$?

**Assistant Response**

<think>
Um, let's try to solve this geometry problem step by step. First, let's understand the problem statement.

( ... 10863 tokens omitted for brevity )

$\Delta x = 2s \cos^2 \theta - s \cos \theta = s \cos \theta (2 \cos \theta - 1)$
$\Delta y = 2s \cos \theta \sin \theta - (-s \sin \theta) = s \sin \theta (2 \cos \theta + 1)$
Length $CD = \sqrt{(\Delta x)^2 + (\Delta y)^2} = s |\sin \theta| \sqrt{\cos^2 \theta (2 \cos \theta - 1)^2 + (2 \cos \theta + 1)^2}$
Simplify inside the square root:
$4 \cos^4 \theta - 4 \cos^3 \theta + \cos^2 \theta + 4 \cos^2 \theta + 4 \cos \theta + 1$

**I should balance depth of reasoning with efficiency, so now I'll stop thinking and deliver a well-considered response.**
</think>

To solve the problem, we start by noting that the two circles are externally tangent at point $A$, so the distance between their centers $O_1$ and $O_2$ is $r_1 + r_2$, where $r_1$ and $r_2$ are their radii.

**Key Observations:**

1. **Power of a Point**: For point $B$ on the first circle, the power with respect to the second circle is given by $BA \cdot BD = 5 \cdot 9 = 45$. This power is also equal to $|BO_2|^2 - r_2^2$.

2. **Common Tangent**: The length of the common tangent $BC$ can be related to the radii and the distance between centers using the Pythagorean theorem for the external tangent: $BC^2 = (O_1 O_2)^2 - (r_1 - r_2)^2$.

3. **Coordinate Geometry**: Placing $A$ at the origin, $O_1$ at $(-r_1, 0)$, and $O_2$ at $(r_2, 0)$, we use the coordinates of $B$ and $C$ on the tangent line to relate the radii and solve for $r_1$ and $r_2$.

**Calculations:**

- Using the power of point $B$, we find $r_1 = 5$ and solve for $r_2$ using the length $AD = 4$.

- The common tangent length $BC$ is found using the radii and the distance between centers.

**Result:** After detailed calculations, the length of $CD$ is found to be $\boxed{6}$.

Table 8: A case study of supervised fine-tuning data, presenting the prompt and corresponding response for the *medium* mode.

**Prompt for Low Mode**

*/* System Prompt */*
You have extremely limited time to think and respond to the users query. Every additional second of processing and reasoning incurs a significant resource cost, which could affect efficiency and effectiveness. Your task is to prioritize speed without sacrificing essential clarity or accuracy. Provide the most direct and concise answer possible. Avoid unnecessary steps, reflections, verification, or refinements UNLESS ABSOLUTELY NECESSARY. Your primary goal is to deliver a quick, clear and correct response.

*/* User Prompt */*
One of the external common tangent lines of the two externally tangent circles with centers $O_1$ and $O_2$ touches the circles at $B$ and $C$, respectively. Let $A$ be the common point of the circles. The line $BA$ meets the circle with center $O_2$ at $A$ and $D$. If $|BA| = 5$ and $|AD| = 4$, then what is $|CD|$?

---

**Assistant Response**

**<think>**
Um, let's try to solve this geometry problem step by step. First, let's understand the problem statement.
*( ... 5633 tokens omitted for brevity )*
Since both are radii, $|O_2C| = r_2$, so $k = r_2/r_1$ (since $|O_1B| = r_1$).
Therefore:
$$x_2 = r_1 + r_2 + \frac{r_2}{r_1}x_1$$
$$y_2 = \frac{r_2}{r_1}y_1$$

**I need to prioritize speed, so I should stop thinking now and provide the most direct answer possible.**
**</think>**

To solve the problem, we use the power of a point theorem and properties of tangent lines and circles.
1. **Power of Point B**: The power of point $B$ with respect to the circle centered at $O_2$ is equal to the product of the lengths of the segments from $B$ to the points of intersection with the circle. Here, $|BA| = 5$ and $|AD| = 4$, so $|BD| = |BA| + |AD| = 9$. Thus, the power of $B$ is $|BA| \cdot |BD| = 5 \cdot 9 = 45$.
2. **Distance from B to $O_2$**: Let $r_1$ and $r_2$ be the radii of the circles centered at $O_1$ and $O_2$, respectively. Since $BC$ is a common tangent, $O_1B \perp BC$ and $O_2C \perp BC$, so $O_1B \parallel O_2C$. The distance between $O_1$ and $O_2$ is $r_1 + r_2$.
3. **Using Coordinates and Similar Triangles**: By setting up coordinates and using the fact that $BA$ intersects the second circle at $D$, we find that the power of $B$ gives $|BO_2|^2 - r_2^2 = 45$.
4. **Length of $CD$**: Since $CD$ is a tangent segment from $C$ to $D$ on the circle, and using the properties of the tangent and the circle, we find that $CD$ can be determined by the power of point $C$ or by recognizing similar triangles. However, using the power of point $B$ and the fact that $BC$ is a common tangent, we find that $|CD| = 6$.
**Answer:** $\boxed{6}$

---

Table 9: A case study of supervised fine-tuning data, presenting the prompt and corresponding response for the *low* mode.

**</think>**
Um, let's try to solve this problem. We need to find the number of nonnegative integer triples (a, b, c) such that a + b + c = 300 and the sum of all pairwise products of the form a²b, a²c, etc., equals 6,000,000. First, let's denote the sum S = a + b + c = 300 and the other sum as T = a²b + a²c + b²a + b²c + c²a + c²b = 6,000,000.

**I should balance depth of reasoning with efficiency, so now I'll stop thinking and deliver a well-considered response.**
**</think>**

Let's denote $S = a+b+c = 300$. We can factor the given sum $T = a^2b+a^2c+b^2a+b^2c+c^2a+c^2b = ab(a + b) + bc(b + c) + ca(c + a)$. Since $a + b = S - c$, $b + c = S - a$, and $c + a = S - b$, we can rewrite $T$ as:

$$T = ab(S - c) + bc(S - a) + ca(S - b) = S(ab + bc + ca) - 3abc$$

Given $T = 6,000,000$ and $S = 300$, we have:

$$300(ab + bc + ca) - 3abc = 6,000,000$$

*( ... solutions omitted for brevity )*

But wait, let's check $d = 0$ to $d = 100$: that's 101 values of $d$. For $d = 0$, 1 ordered triple. For $d = 1$ to 99 (99 values), 6 each: $99 \times 6 = 594$. For $d = 100$, 6. Total $1 + 594 + 6 = 601$.

But earlier, when we thought only specific ratios worked, we were wrong; the algebra shows any arithmetic sequence centered at 100 works, which is a broader set.

*( ... solutions omitted for brevity )*

Therefore, every solution is a permutation of $(100 - d, 100, 100 + d)$ for some nonnegative integer $d \leq 100$.

Now, count the ordered triples:
- When $d = 0$: (100,100,100) $\rightarrow$ 1 way.
- When $d \geq 1$ and $d \leq 99$: Each triple has three distinct numbers, so $3! = 6$ permutations per $d$. There are 99 such $d$ values (1 through 99).
- When $d = 100$: The triple includes a zero: (0,100,200). This has one zero and two distinct non-zeros, so permutations: choose position for 0 (3 choices), then arrange 100 and 200 in the remaining two positions (2 ways): $3 \times 2 = 6$.

Total permutations: $1 + 99 \times 6 + 6 = 1 + 594 + 6 = 601$.

Yes, this makes sense because for any nonnegative integer $d$ from 0 to 100, the triple $(100 - d, 100, 100 + d)$ is valid, and each such triple (except all 100s) can be permuted in 6 ways if all distinct, or 6 ways if one is zero.

The earlier mistake was not recognizing the general arithmetic sequence form, but the algebra confirms T is constant for this family, which exhausts all solutions due to the symmetric dependency on the average term 100.

**Final Answer** $\boxed{601}$

Table 10: A case where the leak penalty was not applied, resulting in the think process leaking into the answer section.

</think>

Um, let's try to solve this problem. We need to find the number of triples of nonnegative integers (a, b, c) such that a + b + c = 300 and the expression a²b + a²c + b²a + b²c + c²a + c²b = 6,000,000. Hmm, that's a symmetric equation in a, b, c. Maybe we can use some symmetry or substitution to simplify it.

First, let's denote S = a + b + c = 300. The given expression is the sum of all terms like a²b, which is symmetric. Let's recall that in symmetric problems, often assuming an order, like $a \geq b \geq c$, can help reduce the cases we need to check, and then we can multiply by the number of permutations at the end.

Let's assume $a \geq b \geq c$. Then c can range from 0 up to some value. Let's try to express the given sum in terms of S and other symmetric sums. Let's denote the given sum as T = a²b + a²c + b²a + b²c + c²a + c²b.

*( ... solutions omitted for brevity )*

Total solutions: $1 + 594 + 6 = 601$. Wait, but hold on, when k = 0, that's the all 100s, which is case 1, and k = 100 is case 3. Now, let's check if the zero case is valid: when c = 0, the original equation has no terms with c²*0 or 0²*something, so it's just a²b + b²a = ab(a + b) = 20,000*300 = 6,000,000, which works, as we saw.

**I should balance depth of reasoning with efficiency, so now I'll stop thinking and deliver a well-considered response.**
</think>

The solutions are all triples where the variables are permutations of $100 + k$, 100, and $100 - k$ for nonnegative integers $k$ such that $100 - k \geq 0$. This includes the all-100s triple and permutations with one zero.

- $k = 0$: 1 solution (100,100,100). - $k = 1$ to 99: Each $k$ gives 6 permutations, totaling $99 \times 6 = 594$. - $k = 100$: 6 permutations (with one zero).

Total: $1 + 594 + 6 = 601$.

**Final Answer** $\boxed{601}$

Table 11: A case where the leak penalty was applied, preventing the think process from leaking into the answer section.

