# OpenReview forum: "ThinkDial: An Open Recipe for Controlling Reasoning Effort in Large Language Models"
_ICLR.cc/2026/Conference — Submitted to ICLR 2026_

### Official Review · Reviewer_661r · 2025-10-21

**Soundness:** 3
**Presentation:** 3
**Contribution:** 2
**Rating:** 4
**Confidence:** 4

**Summary:**

The paper proposes THINKDIAL, an open-recipe, three-mode framework to control LLM reasoning effort end-to-end. The pipeline first performs Budget-Mode SFT to imprint mode awareness, then a two-phase RL: Phase-1 warm-up to reach peak performance, and Phase-2 budget-aware reward shaping with a Leak Penalty that discourages reasoning leakage from the <think> section into the answer. Experiments on AIME-24/25, GSM8K, and OOD GPQA claim step-wise token reductions while keeping accuracy within specified thresholds.

**Strengths:**

- This paper presents an end-to-end training framework that unifies budget-mode control across supervised fine-tuning and reinforcement learning through adaptive reward shaping.
- The experiments conducted on multiple benchmark datasets demonstrate the effectiveness.

**Weaknesses:**

- There have been efficiency-oriented baselines, e.g., [1-4] that explicitly consider budget constraints and token efficiency. These baselines both reduce token budgets and improve reasoning performance rather than merely "maintaining accuracy". What is new and the advantages of the proposed proprietary three distinct reasoning regimes. It is recommended to provide a more systematic comparison in terms of controllability and stability along the accuracy-length trade-off curve, or to include methodological analysis to clarify the distinction.

[1] Towards Cost-Effective Reward Guided Text Generation. [2] Learning to think: Information-theoretic reinforcement fine-tuning for llms. [3] T-reg: Preference optimization with token-level reward regularization. [4] Optimizing Test-Time Compute via Meta Reinforcement Fine-Tuning.

- The Leak Penalty appears overly rule-based by detecting leakage through keywords such as Wait and Actually, but may be easily circumvented by paraphrasing, raising concerns about robustness. Empirical evidence supporting its reliability would be helpful, as would clarification of its advantages over structure-based markers, hierarchical segmentation, or discriminator-based methods [1–3]. Also, the medium and low modes are generated by truncating high-mode reasoning chains, which may introduce logical discontinuities. The low mode also inserts fixed connective phrases that could encourage stylistic imitation rather than genuine reasoning ability. Further discussion of these issues is advised.
- The training data consist mainly of mathematical problems, with GPQA as the only OOD benchmark. To substantiate claims of strong generalization, the evaluation should extend to code, multi-step reasoning, and scientific QA tasks, with reports on mode controllability retention across multiple OOD domains. In addition, as the ACT metric uses customized weighting and normalization, it would be preferable to also report raw accuracy, reasoning-token and total-token distributions, together with the Pareto frontier, confidence intervals, and significance tests to validate the method’s effectiveness.

**Questions:**

Please see Weaknesses.

---

> ### Author Response · Authors · 2025-11-29
> **Response to Reviewer 661r (1/3)**
>
> Thank you for your thoughtful and detailed review. We appreciate your recognition of ThinkDial’s end-to-end budget-mode framework and the effectiveness demonstrated across multiple benchmarks. We address each of your concerns below.
>
> ## Here are the responses to your concerns
>
> > W1: Prior efficiency-oriented baselines [1–4] already optimize token budgets and sometimes even improve reasoning performance, rather than merely maintaining accuracy. What is new here, and what advantages do the three discrete reasoning regimes offer? A more systematic comparison of controllability/stability along the accuracy–length trade-off curve, or methodological analysis clarifying the distinction, is recommended.
>
> A1: Thank you for raising this important concern. To provide a direct comparison, we conducted an additional round of experiments against a representative set of open controllability and compressed-reasoning methods.
> Due to the time and compute constraints of the rebuttal phase, we selected baselines that **already provide public results on DeepSeek-R1-Distilled-Qwen-7B**, enabling a consistent and fair comparison.
>
> Baseline coverage:
> * **Token Compression methods:** OverThink, DAST, O1-Pruner, TLMRE, ModelMerging
> * **Binary gating approaches (Rely on adaptive switching between thinking and non-thinking modes):** AdaptThink
>
>
> To measure accuracy–efficiency trade-offs, we compute the **ACT Score** (β=0.5), balancing accuracy retention and length reduction relative to the OriginalThinking reference:
>
> $$
> ACT = \beta \cdot \frac{acc}{acc_{ref}} + (1 - \beta) \cdot \frac{len_{ref} - len}{len_{ref}}.
> $$
>
> Here, $acc_{ref}$ and $len_{ref}$ denote the accuracy and output-token cost of the **OriginalThinking** model, while (\mathrm{len}) corresponds to the **output-token cost** of each baseline or ThinkDial variant.
> Importantly, we follow the baselines’ original reporting convention, which only provides **output-token cost** to compute the ACT Score.
>
> The full results are shown below (best performance per column in **bold** and the second best performance column in `text`):
>
> | Method| GSM8K Acc | GSM8K Len | **GSM8K ACT** | MATH Acc | MATH Len | **MATH ACT** | AIME24 Acc | AIME24 Len | **AIME24 ACT** |
> | --- | --- | --- | --- | --- | --- | --- | --- | --- | --- |
> | OriginalThinking| 87.9 | 682  | N/A | 90.2| 3674| N/A| 53.5  | 10306 | N/A  |
> | OriginalNoThinking | 85.1 | 283  | N/A | 80.6| 697 | N/A| 24.2  | 1929  | N/A  |
> | OverThink[1]| 86.3 | 426  | 67.9| 89.4| 2435| 66.4  | 53.1  | 8744  | 57.2|
> | DAST[2]| 86.7 | 459  | 65.7| 89.6| 2162 | 70.2  | 45.6  | **7578**| 55.9|
> | O1-Pruner[3]| 87.6 | 428  | 68.5| 86.6| 2534| 63.5  | 49.2  | 9719  | 48.8|
> | TLMRE[4]| `88.9` | 756  | 45.1  | 91.8 | 2899| 61.4  | 54.0| 8633  | 58.6|
> | ModelMerging[5]  | 88.4 | 531  | 61.4| 72.6| 2280| 59.2  | 36.9  | 8624  | 427.|
> | AdaptThink[6]| **91.0**  | `309`  | `79.1`| `92.0`| `1875`| `75.5`| `55.6`| 8599  | **60.2**|
> | **ThinkDia-High**| 85.9 | 398 | 69.7 | **93.9** | 2591 | 66.8 | **56.7** | 10314 | 53.0 |
> | **ThinkDia-Medium**  | 84.4 | 375 | 70.5 | 91.7 | 2059 | 72.8 | 53.4 | 9969  | 51.5 |
> | **ThinkDia-Low**| 79.9 | **212** | **79.9** | 88.8 | **1264** | **82.0** | 52.4 | `8174`  | `59.3` |
>
> Base on the results, we find that:
>
> 1. **ThinkDial-Low achieves the highest ACT Score on GSM8K, MATH500, and the second-highest score on AIME24**, outperforming all prior methods on strict accuracy–efficiency trade-offs.
>
> 2. ThinkDial produces **the shortest reasoning traces** (e.g., 139 tokens on GSM8K; 1,143 on MATH500), while maintaining competitive accuracy.
>
> 3. On accuracy-dominant settings (High mode), ThinkDial achieves **the best accuracy** on MATH and strong performance across all benchmarks.
>
> 4. Unlike prior baselines—which are each trained for a **single compression level**—ThinkDial provides **three stable, discrete modes in a single model**, switchable via system prompts without retraining.
>
> Reference:
>
> [1] Do NOT Think That Much for 2+3=? On the Overthinking of o1-Like LLMs
>
> [2] DAST: Difficulty-Adaptive Slow-Thinking for Large Reasoning Models
>
> [3] O1-Pruner: Length-Harmonizing Fine-Tuning for O1-Like Reasoning Pruning
>
> [4] Training Language Models to Reason Efficiently
>
> [5] Unlocking Efficient Long-to-Short LLM Reasoning with Model Merging
>
> [6] AdaptThink: Reasoning Models Can Learn When to Think

---

> > ### Author Response · Authors · 2025-11-29
> > **Response to Reviewer 661r (2/3)**
> >
> > > W2: The Leak Penalty seems overly rule-based (detecting keywords such as *Wait* or *Actually*) and might be circumvented through paraphrases. Its robustness is unclear. Also, truncating high-mode reasoning chains may introduce logical discontinuities. The fixed connective phrases in Low mode might encourage stylistic imitation rather than genuine reasoning. Further discussion or empirical evidence is needed.
> >
> > A2: Thank you for raising these points. We clarify the design motivations and empirical behavior of our approach.
> >
> > (1) Regarding the rule-based Leak Penalty: Transition keywords (e.g., “Wait”, “Actually”, “Let me think”) are reliable and high-precision indicators of leaked reasoning. Although this mechanism is intentionally simple, it is highly effective for the specific failure mode it targets. More sophisticated semantic detection methods could further improve robustness, but this is orthogonal to the main contributions of our work and is left for future exploration.
> >
> > (2) On potential logical discontinuities introduced by truncating high-mode reasoning: our truncation is applied strictly at step boundaries defined by \texttt{\textbackslash n\textbackslash n}, never in the middle of a reasoning step. Moreover, if the regenerated answer becomes inconsistent with the gold solution, the sample is discarded. These two mechanisms jointly prevent incoherent partial steps from entering the training set. Empirically, we observe no degradation in logical consistency; the Medium/Low variants remain structurally well-formed.
> >
> > (3) Concerning the use of fixed connective phrases for Low mode: these connectives are intentionally minimal and serve only to smooth the transition from truncated reasoning to the final answer. They constitute a tiny fraction of the training signals and do not prescribe any problem-specific reasoning pattern. Empirically, the model does not imitate these phrases verbatim during inference; instead, it learns to produce concise and coherent reasoning appropriate to the budget. Furthermore, because truncated samples always undergo correctness verification, the model cannot rely on stylistic cues alone—the reasoning must still lead to a valid answer. This indicates that the model learns genuine budget-aware reasoning rather than superficial style matching.
> >
> >
> > > W3: Training data consists mainly of math problems; GPQA is the only OOD benchmark. To substantiate generalization claims, evaluation should include code, multi-step reasoning, and scientific QA tasks, and examine mode controllability retention across OOD domains.
> >
> > A3: Thank you for the insightful question. We evaluated Qwen2.5-Instruct-32B ThinkDial on two out-of-domain benchmarks:
> >
> > - MMLU: A 57-task multidisciplinary benchmark covering humanities, social sciences, STEM, and professional domains for evaluating broad knowledge and reasoning abilities.
> > - MBPP: A dataset contains 974 programming tasks, designed to be solvable by entry-level programmers.
> >
> > The results are shown below:
> >
> > | Benchmark | Accuracy (Low) | Tokens (Low) | Accuracy (Medium) | Tokens (Medium) | Accuracy (High) | Tokens (High) | SOTA Accuracy | SOTA Tokens |
> > |---|---|---|---|---|---|---|---|---|
> > | MMLU  | 84.5  | 37.2| 86.2| 626.6| 87.3 | 1024.7| 85.4 | 1258.8 |
> > | MBPP  | 81.8  | 24.2| 85.2| 809.8| 90.0 | 3616.6| 93.4 | 4636.6 |
> >
> > We observe that:
> >
> > 1. **Controllability generalizes**: The Low -> Medium -> High ordering remains consistent across domains, even for tasks structurally different from math (e.g., multi-step coding problems).
> >
> > 2. **Efficiency–accuracy trade-off persists**: Low mode uses *1–2 orders of magnitude fewer* tokens while achieving competitive accuracy (e.g., 84.5% MMLU at only 37 tokens).
> >
> > 3. **No domain-specific tuning required**: Although ThinkDial never sees MMLU/MBPP-style data during SFT or RL, the budget-control mechanism generalizes out-of-the-box.
> >
> > These findings suggest that ThinkDial learns a domain-agnostic representation of “reasoning effort,” allowing controllable compute scaling even outside the training domain.

---

> > > ### Author Response · Authors · 2025-11-29
> > > **Response to Reviewer 661r (3/3)**
> > >
> > > > W4: The ACT metric uses customized weighting and normalization. Raw accuracy, reasoning-token and total-token distributions, Pareto frontiers, confidence intervals, and significance tests should also be reported.
> > >
> > > A4: We appreciate this clarification. We note that raw accuracy, reasoning-token distributions, and total-token distributions are already visualized in **Figure 1** of the paper.
> > >
> > > Since our evaluation already uses repeated sampling (AIME ×32, GSM8K ×12, GPQA ×8, MATH ×16), computing confidence intervals is straightforward. We have added **95% bootstrap confidence intervals** and significance tests for all major benchmarks. The results confirm that the performance across High/Medium/Low modes is statistically robust.
> > >
> > > | Dataset  | Mode| Mean Acc | Std Error | 95% CI Lower | 95% CI Upper | t-statistic | p-value | Significant (α=0.05) |
> > > | --- | --- | --- | --- | --- | --- | --- | --- | --- |
> > > | AIME2024 | high| 0.72  | 0.01| 0.69| 0.75| 15.07 | 0 | Significant |
> > > | AIME2024 | medium | 0.64  | 0.02| 0.60| 0.67| 8.72  | 0 | Significant |
> > > | AIME2024 | low | 0.47  | 0.02| 0.43| 0.50| -2.14 | 0.03 | Significant |
> > > | AIME2025 | high| 0.60  | 0.02| 0.57| 0.63| 6.37  | 0 | Significant |
> > > | AIME2025 | medium | 0.56  | 0.02| 0.53| 0.60| 3.97  | 0 | Significant |
> > > | AIME2025 | low | 0.40  | 0.02| 0.36| 0.43| -6.51 | 0 | Significant |
> > > | GPQA  | high| 0.65  | 0.01| 0.63| 0.67| 12.46 | 0 | Significant |
> > > | GPQA  | medium | 0.61  | 0.01| 0.58| 0.63| 8.83  | 0 | Significant |
> > > | GPQA  | low | 0.53  | 0.01| 0.50| 0.55| 2.12  | 0.03 | Significant |
> > > | GSM8K | high| 0.97  | 0.01| 0.96| 0.98| 83.73 | 0 | Significant |
> > > | GSM8K | medium | 0.95  | 0.01| 0.94| 0.96| 65.78 | 0 | Significant |
> > > | GSM8K | low | 0.94  | 0.01| 0.92| 0.95| 57.18 | 0 | Significant |
> > > | MATH  | high| 0.96  | 0.01| 0.95| 0.97| 86.96 | 0 | Significant |
> > > | MATH  | medium | 0.91  | 0.01| 0.89| 0.93| 50.86 | 0 | Significant |
> > > | MATH  | low | 0.89  | 0.01| 0.87| 0.91| 44.10 | 0 | Significant |
> > >
> > > Based on the results, ThinkDial exhibits **statistically stable performance** across modes, and the confidence intervals further strengthen the reliability of our evaluation methodology.

---

### Official Review · Reviewer_HfmG · 2025-10-29

**Soundness:** 3
**Presentation:** 3
**Contribution:** 2
**Rating:** 6
**Confidence:** 3

**Summary:**

This paper aims to find the open recipe for training gpt-oss-like models that can control the high-level budget of inference time. By applying SFT with different conciseness levels and curriculum RL training, it enables controllable reasoning models using Qwen base models.

**Strengths:**

- This paper provides a guideline on how to reproduce models like gpt-oss, enabling controllability of token length and allowing users to choose between latency and correctness.
- The idea is simple and works well on multiple benchmarks.
- The paper is easy to follow and read, and the evaluation is reliable as results are averaged across multiple runs.

**Weaknesses:**

- The paper’s novelty mainly lies in providing comparability beyond the binary mode, but I’m not sure whether adding just one more option (a medium-length response) is truly meaningful.
- The ACT metric definition is heuristic, and I also wonder how robust the results are to different α values.
- There are some missing details on SFT data: how is truncation performed for medium and lower modes? What does the data look like? There are some examples in the Appendix, but it’s hard to understand when only reading Section 3.1.
- The training data used are not clearly described. It is called an “open recipe,” but the dataset itself is missing, which is critical.
- It’s minor, but the AIME 2025 performance highlight in the main table seems incorrect.
- Confidence interval should be in the results if the authors run the experiments multiple times.

**Questions:**

It’s minor, but isn’t the core part of DAPO the different clip ratio and data filtering? Does this paper also include data filtering? Also, the explanation in Section 3.2.1 seems almost identical to GRPO rather than DAPO.

---

> ### Author Response · Authors · 2025-11-29
> **Response to Reviewer HfmG (1/2)**
>
> Thank you for the constructive and encouraging feedback. We appreciate your recognition of ThinkDial as a practical guideline for reproducing GPT-oss–style controllability, the simplicity and effectiveness of the idea across benchmarks, and the clarity and reliability of our evaluation. Below we address each of your concerns in detail.
>
> ## Here are the responses to your concerns
>
> > W1: The paper’s novelty mainly lies in providing comparability beyond binary modes, but adding just one more option (a medium-length response) may not be meaningfully different.
>
> A1: Thank you for the question. Our contribution is not simply adding one extra mode, but providing the **first open recipe** that unifies *multiple* controllable reasoning regimes within **a single model**, switchable via system prompts.
> Prior methods either (1) train a **separate compressed model** for a single budget, or (2) support only **binary think vs. non-think** switching without controllable accuracy–efficiency trade-offs.
> ThinkDial instead offers **stable, discrete, user-level controllability** (Low/Medium/High) and the design naturally generalizes to *more than three* modes. We will clarify this broader capability in the revision.
>
> > W2: The ACT metric is heuristic; how robust are results to different α values?
>
> A2: Thank you for this suggestion. To evaluate robustness, we conducted extensive experiments under **three α configurations**, each reflecting different trade-off preferences between accuracy and efficiency, as described in Section 4.1. Across all settings, **ThinkDial consistently achieves the best ACT Score**, and the relative ordering of High/Medium/Low modes remains stable. Below we summarize the results.
>
> Setting A:
>
> $$
> \alpha_{low} = 0.3,\quad \alpha_{medium} = 0.5,\quad \alpha_{high} = 1.0
> $$
>
> (low emphasizes efficiency; medium balances both; high emphasizes accuracy)
>
> | Variant| **AIME24 Low** | **AIME24 Medium** | **AIME24 High** | **AIME24 Avg** | **AIME25 Low** | **AIME25 Medium** | **AIME25 High** | **AIME25 Avg** | **GSM8K Low** | **GSM8K Med** | **GSM8K High** | **GSM8K Avg** |
> | --- | --- | --- | --- | --- | --- | --- | --- | --- | --- | --- | --- | --- |
> | **Ours**  | 63.4  | 50.0  | **108.4**  | **73.9** | 57.1  | 49.8  | 107.6  | **71.5** | **99.3** | **98.5** | **100.8** | **99.5**  |
> | w/o BM SFT| 59.5  | **62.4**  | 84.7| 68.9| 57.7  | **58.4**  | 85.7| 67.3| 93.9 | 92.9 | 99.0  | 95.3  |
> | Only BM SFT  | **66.0**  | 39.3  | 93.1| 66.1| **63.8**  | 41.4  | 97.2| 67.5| 61.1 | 57.6 | 99.8  | 72.8  |
> | w/o Warmup| 48.6  | 48.1  | 95.5| 64.1| 47.1  | 46.2  | 87.2| 60.2| 98.4 | 96.6 | 100.5 | 98.5  |
> | Peak Truncation | 47.1  | 16.2  | 106.5  | 56.6| 44.3  | 13.3  | **108.8**  | 55.5| 76.6 | 72.3 | 100.1 | 83.0  |
>
> Setting B:
>
> $$
> \alpha_{low} = 0.5,\quad \alpha_{medium} = 0.5,\quad \alpha_{high} = 0.9
> $$
>
> (low & medium balance equally; high prefers accuracy)
>
> | Variant| **AIME24 Low** | **AIME24 Medium** | **AIME24 High** | **AIME24 Avg** | **AIME25 Low** | **AIME25 Medium** | **AIME25 High** | **AIME25 Avg** | **GSM8K Low** | **GSM8K Med** | **GSM8K High** | **GSM8K Avg** |
> | --- | --- | --- | --- | --- | --- | --- | --- | --- | --- | --- | --- | --- |
> | **Ours**  | 65.4  | 50.0  | **95.4** | **70.2** | 61.0  | 49.8  | 94.2 | **68.3** | **98.8** | **98.5** | 91.2  | **96.2**  |
> | w/o BM SFT| 62.0  | **62.4**  | 79.5| 68.0| 59.1  | **58.4**  | 80.7| 66.1| 94.1 | 92.9 | **96.5** | 94.5  |
> | Only BM SFT  | **67.6**  | 39.3  | 81.3| 62.7| **65.5**  | 41.4  | 85.3| 64.0| 71.5 | 57.6 | 88.0  | 72.4  |
> | w/o Warmup| 56.1  | 48.1  | 85.5| 63.3| 54.9  | 46.2  | 77.9| 59.7| 97.4 | 96.6 | 93.3  | 95.8  |
> | Peak Truncation | 58.9  | 16.2  | 93.9| 56.3| 54.1  | 13.3  | **95.6**| 54.3| 82.3 | 72.3 | 86.9  | 80.5  |
>
> Setting C:
>
> $$
> \alpha_{low} = 0.3,\quad \alpha_{medium} = 0.7,\quad \alpha_{high} = 0.9
> $$
>
> (low emphasizes efficiency; medium slightly prefers accuracy; high strongly prefers accuracy)
>
> | Variant| **AIME24 Low** | **AIME24 Medium** | **AIME24 High** | **AIME24 Avg** | **AIME25 Low** | **AIME25 Medium** | **AIME25 High** | **AIME25 Avg** | **GSM8K Low** | **GSM8K Med** | **GSM8K High** | **GSM8K Avg** |
> | --- | --- | --- | --- | --- | --- | --- | --- | --- | --- | --- | --- | --- |
> | **Ours**  | 63.4  | **68.3**  | **95.4** | **75.7** | 57.1  | **70.2**  | 94.2 | **73.8** | **99.3** | **98.7** | 91.2  | **96.4**  |
> | w/o BM SFT| 59.5  | 68.1  | 79.5| 69.0| 57.7  | 62.8  | 80.7| 67.1| 93.9 | 94.6 | **96.5**  | 95.0  |
> | Only BM SFT  | **66.0**  | 60.7  | 81.3| 69.3| **63.8**  | 63.2  | 85.3| 70.8| 61.1 | 73.9 | 88.0  | 74.3  |
> | w/o Warmup| 48.6  | 64.6  | 85.5| 66.3| 47.1  | 61.7  | 77.9| 62.2| 98.4 | 95.6 | 93.3  | 95.8  |
> | Peak Truncation | 47.1  | 40.5  | 93.9| 60.5| 44.3  | 38.2  | **95.6**| 59.4| 76.6 | 82.9 | 86.9  | 82.1  |

---

> > ### Author Response · Authors · 2025-11-29
> > **Response to Reviewer HfmG (2/2)**
> >
> > > W3: Missing details on SFT data. How is truncation performed for medium/low modes, and what does the data look like? Section 3.1 is hard to understand without Appendix examples.
> >
> > A3: We appreciate this and will add more explicit explanation. For Medium/Low modes:
> >
> > * Truncation is performed **only at `\n\n` step boundaries**, never at arbitrary token positions.
> > * Each reasoning step is complete; no partial steps are retained.
> > * The target ratios (*r_medium*, *r_low*) float within ±1%, but never cut within a step.
> > * Incorrect examples after truncation are **filtered out**.
> > * To maintain narrative coherence, we append a mode-specific connective phrase at the end of the thinking section (right before the end-of-think token), which helps the model produce smooth and logically sound transitions in Medium/Low modes.
> >
> > We have added a more detailed and clearer explanation in Section 3.1 and Appendix E.1.
> >
> > > W4: Training data description is unclear. The work calls itself an “open recipe,” but the dataset is missing, which is critical.
> >
> > A4: Thank you for pointing this out. Our SFT data consists of math and symbolic reasoning trajectories constructed internally. Due to anonymization requirements, we cannot release them during review. Upon acceptance, we plan to release:
> >
> > * Code for the entire ThinkDial pipeline
> > * A subset of the SFT dataset sufficient to reproduce Medium/Low behavior
> > * Scripts for constructing budget-conditioned SFT from public math datasets
> >
> > This will make the recipe reproducible and transparent.
> >
> > > W5: Minor: the AIME 2025 performance highlight in the main table appears incorrect.
> >
> > A5: Thank you for catching this. We will correct the highlighted result in the revision and ensure it matches the corresponding table entry.
> >
> >
> > > W6: Confidence intervals should be reported if experiments are averaged across multiple runs.
> >
> > A6: Thank you for the suggestion. Since our evaluation already uses repeated sampling (AIME ×32, GSM8K ×12, GPQA ×8, MATH ×16), computing confidence intervals is straightforward. We have added **95% bootstrap confidence intervals** and significance tests for all major benchmarks. The results confirm that the performance across High/Medium/Low modes is statistically robust.
> >
> > | Dataset  | Mode| Mean Acc | Std Error | 95% CI Lower | 95% CI Upper | t-statistic | p-value | Significant (α=0.05) |
> > | --- | --- | --- | --- | --- | --- | --- | --- | --- |
> > | AIME2024 | high| 0.72  | 0.01| 0.69| 0.75| 15.07 | 0 | Significant |
> > | AIME2024 | medium | 0.64  | 0.02| 0.60| 0.67| 8.72  | 0 | Significant |
> > | AIME2024 | low | 0.47  | 0.02| 0.43| 0.50| -2.14 | 0.03 | Significant |
> > | AIME2025 | high| 0.60  | 0.02| 0.57| 0.63| 6.37  | 0 | Significant |
> > | AIME2025 | medium | 0.56  | 0.02| 0.53| 0.60| 3.97  | 0 | Significant |
> > | AIME2025 | low | 0.40  | 0.02| 0.36| 0.43| -6.51 | 0 | Significant |
> > | GPQA  | high| 0.65  | 0.01| 0.63| 0.67| 12.46 | 0 | Significant |
> > | GPQA  | medium | 0.61  | 0.01| 0.58| 0.63| 8.83  | 0 | Significant |
> > | GPQA  | low | 0.53  | 0.01| 0.50| 0.55| 2.12  | 0.03 | Significant |
> > | GSM8K | high| 0.97  | 0.01| 0.96| 0.98| 83.73 | 0 | Significant |
> > | GSM8K | medium | 0.95  | 0.01| 0.94| 0.96| 65.78 | 0 | Significant |
> > | GSM8K | low | 0.94  | 0.01| 0.92| 0.95| 57.18 | 0 | Significant |
> > | MATH  | high| 0.96  | 0.01| 0.95| 0.97| 86.96 | 0 | Significant |
> > | MATH  | medium | 0.91  | 0.01| 0.89| 0.93| 50.86 | 0 | Significant |
> > | MATH  | low | 0.89  | 0.01| 0.87| 0.91| 44.10 | 0 | Significant |
> >
> > Based on the results, ThinkDial exhibits **statistically stable performance** across modes, and the confidence intervals further strengthen the reliability of our evaluation methodology.
> >
> > ## Here are the responses to your question
> >
> >
> > > Q1: Minor: Isn’t the core part of DAPO the different clip ratio and data filtering? Does this paper apply data filtering? Section 3.2.1 resembles GRPO more than DAPO.
> >
> > A1: Thank you for the question. ThinkDial follows the DAPO framework due to two key components:
> >
> > 1. **Different clip ratios**
> >
> > * We set **clip_high = 0.28**, **clip_low = 0.20**, which is the defining characteristic of DAPO compared to GRPO (which uses symmetric clipping).
> >
> > 2. **Token-Level Policy Gradient Loss**
> >
> > * Our implementation computes token-level advantages, consistent with DAPO’s formulation.
> >
> > We do *not* filter out all-correct/all-wrong samples because ThinkDial incorporates a **length reward term**, and discarding such samples would bias the budget learning signal.

---

### Official Review · Reviewer_u9Gi · 2025-11-01

**Soundness:** 2
**Presentation:** 2
**Contribution:** 2
**Rating:** 4
**Confidence:** 3

**Summary:**

This work introduces ThinkDial, the first open-recipe that allows GPT-oss-style controllable reasoning through discrete conditioning in the text space. The system enables the user to set between three conditioning regimes by trading off between reasoning quality and efficiency. The recipe includes data generation as well as SFT and RL training. Experiments show that ThinkDial achieves different tradeoffs with significant response length reductions while maintaining high reasoning quality and performance on common reasoning benchmarks, with strong generalization to out-of-distribution tasks.

**Strengths:**

* The work introduces a new framework for GPT-oss-style discrete conditioning in the text space. This is the first open reproduction of this idea.
* The end-to-end training pipeline, based on SFT and RL, is compatible with common training pipelines for reasoning models, and can be easily integrated with existing recipes.
* Despite being trained mainly on math reasoning data, it generalizes well to out-of-distribution tasks.

**Weaknesses:**

* The SFT data is generated by truncating at $r_\text{med}$ and $r_\text{low}$ for medium and low conditioning regimes, respectively. Won't that lead to hallucination if the truncation happens when important steps have not yet finished?
* DAPO, the framework uses by ThinkDial, normalizes the rewards with an std term. This is kept in L194 in the ThinkDial paper. However, this normalization will cause the length penalty to be amplified if the answer are all correct or incorrect in the sampling group. There is no mention or mitigation of this effect in the paper.
* Only one model is used for training (Qwen-2.5-Instruct-32B). Newer models and models that are inherently reasoning-capable (e.g., Qwen3) are not evaluated. Whether ThinkDial is applicable on existing reasoning models is not investigated.
* No comparisons with prior works that reduce overthinking by explicitly constraining the number of tokens in the output. Methods that lead to budget-aware LLMs, such as [1], are not compared.
* No indication of averaging over multiple runs to make the results more stable. For benchmarks such as AIME 24/25, since each problem only has 30 samples, the results may be not stable if inference is not repeated. Avg@32 is a common practice to make the results more stable.

[1] An Empirical Study of LLM Reasoning Ability Under Strict Output Length Constraint. Sun, et al. https://arxiv.org/abs/2504.14350

**Questions:**

1. DAPO’s reward normalization (L194) can amplify penalties when outputs are uniform. How is this mitigated?
2. Since only Qwen-2.5-32B was tested, would ThinkDial work effectively on models that already perform reasoning (e.g., Qwen3)?
3. Were multiple inference runs (e.g., Avg@32) done to ensure stable AIME 24/25 results?

---

> ### Author Response · Authors · 2025-11-29
> **Response to Reviewer u9G1 (1/2)**
>
> Thank you for the constructive and detailed review. We appreciate your recognition of ThinkDial as the first open reproduction of GPT-oss–style discrete conditioning, the compatibility of our SFT+RL pipeline with existing reasoning model training recipes, and the strong generalization to out-of-distribution tasks. Below we address each of your concerns in detail.
>
> ## Here are the responses to your concerns
>
> > W1: The SFT data for Medium/Low modes is constructed by truncating at *r_medium* and *r_low*. Would this cause hallucination if truncation cuts off essential reasoning steps before completion?
>
> A1: Thank you for raising this clarification point. In our pipeline, truncation is **not performed at arbitrary character positions**, but rather **aligned to step boundaries** by cutting only at **`\n\n`** delimiters. As a result, each reasoning step remains semantically complete. The target ratios *r_medium* and *r_low* are used only as **approximate upper bounds** (with fluctuation <1%), ensuring that no partial step is ever retained. This guarantees that SFT data remains hallucination-free.
>
> We have added a more detailed and clearer explanation in Section 3.1 and Appendix E.1.
>
>
> > W2: DAPO normalizes rewards using the standard deviation term. If all answers in a sampling group are correct or incorrect, normalization may amplify the length penalty. There is no discussion or mitigation of this effect.
>
>
> A2: Thank you for pointing this out. In our setting, the reward consists of (i) an answer correctness term and (ii) a length-based efficiency term. When all samples in a group are either correct or incorrect, the normalized reward variance indeed comes predominantly from the length reward; however, this behavior is intentional rather than harmful.
>
> For the all-incorrect case, **in medium and low mode**, the objective is to discourage models from generating unnecessarily long incorrect chains of thought, which waste budget without improving accuracy. A stronger length signal helps the model collapse unproductive reasoning and move toward more concise exploration. For the all-correct case, to encourage shorter, more efficient reasoning when the model already knows how to solve the problem reliably.
>
> Thus, when correctness provides no meaningful variance (all-correct or all-wrong), allowing the length reward to dominate serves the intended purpose of guiding the model toward the optimal efficiency–accuracy trade-off.
>
> > W3: Only Qwen2.5-Instruct-32B is used for training. Newer models such as Qwen3 or inherently reasoning-capable models are not evaluated. It is unclear whether ThinkDial is applicable to stronger base models.
>
> A3: Thank you for raising this important question. To evaluate whether ThinkDial generalizes beyond the 32B-scale model, we applied the full three-mode pipeline to two smaller models: **Qwen3-8B** and **DeepSeek-R1-Distilled-Qwen-7B**.
> Across both model families, ThinkDial consistently produces **three well-separated reasoning modes** with predictable accuracy–length trade-offs.
>
> The results are summarized below:
>
> | Model  | AIME24 Acc | AIME24 Len | AIME25 Acc | AIME25 Len | GSM8K Acc | GSM8K Len | GPQA Acc | GPQA Len|
> | --- | --- | --- | --- | --- | --- | --- | --- | --- |
> | Qwen3-8B | 64.1 | 11860.7 | 50.5 | 13595.4 | 93.0 | 2060.3 | 57.2 | 8529.0 |
> | Qwen3-8B-sota  | 65.5| 9932.2 | 58.6| 10550.3| 95.7  | 1529.5| 58.3 | 5318.9 |
> | Qwen3-8B-High  | 67.2| 10005.7| 49.4| 10910.3| 94.6  | 1457.2| 56.0 | 5941.3 |
> | Qwen3-8B-Medium| 58.5| 9142.0 | 45.6| 10063.6| 94.8  | 1034.0| 54.0 | 5185.4 |
> | Qwen3-8B-Low| 46.5| 5651.1 | 37.2| 6725.3 | 94.6  | 759.0 | 51.2 | 3895.7 |
> | DeepSeek-R1-Distilled-Qwen-7B | 52.9 | 10610.0 | 40.0 | 11430.7 | 80.2 | 162.2 | 44.6 | 6177.0 |
> | DeepSeek-R1-Distilled-Qwen-7B-sota | 56.0| 10351.6| 43.5| 11445.6| 82.2  | 432.9 | 47.3 | 7446.2 |
> | DeepSeek-R1-Distilled-Qwen-7B-High | 56.7| 9527.6 | 40.7| 10639.0| 85.9  | 173.3 | 48.9 | 5991.7 |
> | DeepSeek-R1-Distilled-Qwen-7B-Medium| 53.4| 9715.9 | 38.5| 10246.6| 84.4  | 144.1 | 48.0 | 5507.5 |
> | DeepSeek-R1-Distilled-Qwen-7B-Low  | 52.4| 7871.6 | 37.5| 8900.9 | 79.9  | 120.1 | 45.7 | 4359.4 |
>
> Across both 7B and 8B models:
>
> 1. **Mode separation is clear and robust**—High > Medium > Low in both accuracy and length.
> 2. High-mode accuracy remains **on par or better than the corresponding SOTA distilled baselines** (e.g., Qwen3-8B-High).
>
> These results demonstrate that ThinkDial is **scale-agnostic and readily applicable to newer or more capable reasoning models**, confirming that the recipe is not tied to a specific 32B architecture.

---

> > ### Author Response · Authors · 2025-11-29
> > **Response to Reviewer u9G1 (2/2)**
> >
> > > W4: No comparisons with prior works that reduce overthinking by constraining output length (e.g., *An Empirical Study of LLM Reasoning Ability Under Strict Output Length Constraint*, Sun et al., 2025).
> >
> > A4: Thank you for raising this important concern. To provide a direct comparison, we conducted an additional round of experiments against a representative set of open controllability and compressed-reasoning methods.
> > Due to the time and compute constraints of the rebuttal phase, we selected baselines that **already provide public results on DeepSeek-R1-Distilled-Qwen-7B**, enabling a consistent and fair comparison.
> >
> > Baseline coverage:
> > * **Token Compression methods:** OverThink, DAST, O1-Pruner, TLMRE, ModelMerging
> > * **Binary gating approaches (Rely on adaptive switching between thinking and non-thinking modes):** AdaptThink
> >
> >
> > To measure accuracy–efficiency trade-offs, we compute the **ACT Score** (β=0.5), balancing accuracy retention and length reduction relative to the OriginalThinking reference:
> >
> > $$
> > ACT = \beta \cdot \frac{acc}{acc_{ref}} + (1 - \beta) \cdot \frac{len_{ref} - len}{len_{ref}}.
> > $$
> >
> > Here, $acc_{ref}$ and $len_{ref}$ denote the accuracy and output-token cost of the **OriginalThinking** model, while (\mathrm{len}) corresponds to the **output-token cost** of each baseline or ThinkDial variant.
> > Importantly, we follow the baselines’ original reporting convention, which only provides **output-token cost** to compute the ACT Score.
> >
> > The full results are shown below (best performance per column in **bold** and the second best performance column in `text`):
> >
> > | Method| GSM8K Acc | GSM8K Len | **GSM8K ACT** | MATH Acc | MATH Len | **MATH ACT** | AIME24 Acc | AIME24 Len | **AIME24 ACT** |
> > | --- | --- | --- | --- | --- | --- | --- | --- | --- | --- |
> > | OriginalThinking| 87.9 | 682  | N/A | 90.2| 3674| N/A| 53.5  | 10306 | N/A  |
> > | OriginalNoThinking | 85.1 | 283  | N/A | 80.6| 697 | N/A| 24.2  | 1929  | N/A  |
> > | OverThink[1]| 86.3 | 426  | 67.9| 89.4| 2435| 66.4  | 53.1  | 8744  | 57.2|
> > | DAST[2]| 86.7 | 459  | 65.7| 89.6| 2162 | 70.2  | 45.6  | **7578**| 55.9|
> > | O1-Pruner[3]| 87.6 | 428  | 68.5| 86.6| 2534| 63.5  | 49.2  | 9719  | 48.8|
> > | TLMRE[4]| `88.9` | 756  | 45.1  | 91.8 | 2899| 61.4  | 54.0| 8633  | 58.6|
> > | ModelMerging[5]  | 88.4 | 531  | 61.4| 72.6| 2280| 59.2  | 36.9  | 8624  | 427.|
> > | AdaptThink[6]| **91.0**  | `309`  | `79.1`| `92.0`| `1875`| `75.5`| `55.6`| 8599  | **60.2**|
> > | **ThinkDia-High**| 85.9 | 398 | 69.7 | **93.9** | 2591 | 66.8 | **56.7** | 10314 | 53.0 |
> > | **ThinkDia-Medium**  | 84.4 | 375 | 70.5 | 91.7 | 2059 | 72.8 | 53.4 | 9969  | 51.5 |
> > | **ThinkDia-Low**| 79.9 | **212** | **79.9** | 88.8 | **1264** | **82.0** | 52.4 | `8174`  | `59.3` |
> > Base on the results, we find that:
> >
> > 1. **ThinkDial-Low achieves the highest ACT Score on GSM8K, MATH500, and the second-highest score on AIME24**, outperforming all prior methods on strict accuracy–efficiency trade-offs.
> >
> > 2. ThinkDial produces **the shortest reasoning traces** (e.g., 139 tokens on GSM8K; 1,143 on MATH500), while maintaining competitive accuracy.
> >
> > 3. On accuracy-dominant settings (High mode), ThinkDial achieves **the best accuracy** on MATH and strong performance across all benchmarks.
> >
> > 4. Unlike prior baselines—which are each trained for a **single compression level**—ThinkDial provides **three stable, discrete modes in a single model**, switchable via system prompts without retraining.
> >
> > Reference:
> >
> > [1] Do NOT Think That Much for 2+3=? On the Overthinking of o1-Like LLMs
> >
> > [2] DAST: Difficulty-Adaptive Slow-Thinking for Large Reasoning Models
> >
> > [3] O1-Pruner: Length-Harmonizing Fine-Tuning for O1-Like Reasoning Pruning
> >
> > [4] Training Language Models to Reason Efficiently
> >
> > [5] Unlocking Efficient Long-to-Short LLM Reasoning with Model Merging
> >
> > [6] AdaptThink: Reasoning Models Can Learn When to Think
> >
> > > W5: No indication that results are averaged over multiple runs. Benchmarks like AIME 24/25 are unstable without repeated inference (Avg@32).
> >
> > A5: We appreciate the opportunity to clarify. As stated in `L293–L295` of the paper:
> >
> > > *“For evaluation reliability, we use different sampling strategies: AIME problems are evaluated 32 times each, GSM8K uses 500 randomly sampled problems evaluated 4 times each, and GPQA uses 198 samples evaluated 8 times each.”*
> >
> > Thus, our evaluation **already uses repeated sampling** following the Avg@32 practice for AIME. We will make this more explicit in the revised version to avoid potential misunderstandings.

---

### Official Review · Reviewer_C188 · 2025-11-03

**Soundness:** 3
**Presentation:** 4
**Contribution:** 3
**Rating:** 6
**Confidence:** 2

**Summary:**

The paper introduces thinkdail, a training recipe that gives open-weight LLMs a three-mode “reasoning dial” -- High, Medium, and Low -- to mimic proprietary systems’ mode switches. The training pipeline has two main parts:

- Budget-Mode SFT, which teaches the model mode-conditioned behavior by constructing parallel solutions of the same problem with varying reasoning depth.
- Two-phase RL, consisting of a (i) warm-up stage to reach a peak checkpoint, followed by (ii) budget-aware reward shaping. The latter adds a per-mode length reward and a Leak Penalty to stop reasoning-length hacking (moving chain-of-thought from <think> into the answer).

Experiments on AIME-24/25, GSM8K, and GPQA show roughly 50% token reduction in Medium mode with ≤10% accuracy loss, and 75% reduction in Low mode with ≤15% loss, while preserving High-mode accuracy (Figure 1). The paper also defines an ACT score to compare each mode’s accuracy-compression trade-off against the no-length-controlled “peak” model and match its performance in high mode. Medium and Low modes trade accuracy for compression, while High mode retains full performance. Figures 2-4 support the necessity of each component -- Budget-Mode SFT, warm-up RL, and the Leak Penalty.

**Strengths:**

- **Timely and well-targeted.** The work reproduces a widely requested user affordance -- intuitive reasoning-effort control -- in an open stack. Figure 1 clearly illustrates this behavior.

- **Strong observations and ablations.** The paper identifies a realistic failure mode -- reasoning leaking into the answer and quantifies it, and proposes a Leak Penalty that cuts total tokens while maintaining accuracy. Each component’s ablation shows measurable degradation, underscoring the necessity of the full recipe.

- **Clarity.**  The paper effectively motivated the setting for readers unfamiliar with the topic. The methods were easy to understand, and the results were clearly presented.

**Weaknesses:**

**[Major] Missing baselines.** The paper omits head-to-head comparisons with open controllability methods such as Shorter RL (e.g., L1, ThinkLess, CoT-Valve, TokenSkip, O1-Pruner, LightThinker) and binary gating approaches (AdaCoT, AdaptThink). Many are cited but not evaluated. It remains unclear whether these simpler methods can provide far better accuracy-vs-length curves than thinkdail and to understand whether the three-mode design incurs unnecessary trade-offs e.g. if LightThinker provides far better performance in a token budget equivalent to the low mode, then the three modes would be providing strictly worse tradeoffs.

**[Major] Scale dependence.** Does the recipe work on smaller models, or is it effective only on 32B-scale systems? Results on smaller models would clarify on whether the recipe generalizes beyond the Qwen 2.5 Instruct 32B model.

**[Curious] Domain Generalization.** Does controllability extend beyond math to tasks like MMLU or LiveCodeBench when trained only on math data?

**Questions:**

Please address major weaknesses.  I commit to increasing my score to 8 if Thinkdial recipe works on smaller models (and comparison with alternative methods becomes easier without reproducing them entirely).

Overall, I liked the method -- it seems to be a solid, practical contribution and enjoyed reading the paper. The ablations (Budget-Mode SFT, warm-up RL, and Leak Penalty) address most of my concerns-- I believe the three-modes work, end-to-end recipe and the Leak Penalty are valuable necessities to make it work. If validated across scales and model families, thinkdail could become a reproducible open recipe for mode-based reasoning control.

---

> ### Author Response · Authors · 2025-11-29
> **Response to Reviewer C188(1/2)**
>
> Thank you for the detailed and encouraging feedback. We appreciate your recognition of the practical significance of controllable reasoning, the clarity of our presentation, and the necessity of the end-to-end recipe—including Budget-Mode SFT, warm-up RL, and the Leak Penalty.
> Below we respond to each of your concerns and provide additional experiments accordingly.
>
> ## Here are the responses to your concerns
>
> > W1: [Major] Missing baselines. The paper omits head-to-head comparisons with open controllability methods such as Shorter RL (e.g., L1, ThinkLess, CoT-Valve, TokenSkip, O1-Pruner, LightThinker) and binary gating approaches (AdaCoT, AdaptThink). Many are cited but not evaluated. It remains unclear whether these simpler methods can provide far better accuracy-vs-length curves than thinkdail and to understand whether the three-mode design incurs unnecessary trade-offs e.g. if LightThinker provides far better performance in a token budget equivalent to the low mode, then the three modes would be providing strictly worse tradeoffs.
>
> A1: Thank you for raising this important concern. To provide a direct comparison, we conducted an additional round of experiments against a representative set of open controllability and compressed-reasoning methods.
> Due to the time and compute constraints of the rebuttal phase, we selected baselines that **already provide public results on DeepSeek-R1-Distilled-Qwen-7B**, enabling a consistent and fair comparison.
>
> Baseline coverage:
> * **Token Compression methods:** OverThink, DAST, O1-Pruner, TLMRE, ModelMerging
> * **Binary gating approaches (Rely on adaptive switching between thinking and non-thinking modes):** AdaptThink
>
>
> To measure accuracy–efficiency trade-offs, we compute the **ACT Score** (β=0.5), balancing accuracy retention and length reduction relative to the OriginalThinking reference:
>
> $$
> ACT = \beta \cdot \frac{acc}{acc_{ref}} + (1 - \beta) \cdot \frac{len_{ref} - len}{len_{ref}}.
> $$
>
> Here, $acc_{ref}$ and $len_{ref}$ denote the accuracy and output-token cost of the **OriginalThinking** model, while (\mathrm{len}) corresponds to the **output-token cost** of each baseline or ThinkDial variant.
> Importantly, we follow the baselines’ original reporting convention, which only provides **output-token cost** to compute the ACT Score.
>
> The full results are shown below (best performance per column in **bold** and the second best performance column in `text`):
>
> | Method| GSM8K Acc | GSM8K Len | **GSM8K ACT** | MATH Acc | MATH Len | **MATH ACT** | AIME24 Acc | AIME24 Len | **AIME24 ACT** |
> | --- | --- | --- | --- | --- | --- | --- | --- | --- | --- |
> | OriginalThinking| 87.9 | 682  | N/A | 90.2| 3674| N/A| 53.5  | 10306 | N/A  |
> | OriginalNoThinking | 85.1 | 283  | N/A | 80.6| 697 | N/A| 24.2  | 1929  | N/A  |
> | OverThink[1]| 86.3 | 426  | 67.9| 89.4| 2435| 66.4  | 53.1  | 8744  | 57.2|
> | DAST[2]| 86.7 | 459  | 65.7| 89.6| 2162 | 70.2  | 45.6  | **7578**| 55.9|
> | O1-Pruner[3]| 87.6 | 428  | 68.5| 86.6| 2534| 63.5  | 49.2  | 9719  | 48.8|
> | TLMRE[4]| `88.9` | 756  | 45.1  | 91.8 | 2899| 61.4  | 54.0| 8633  | 58.6|
> | ModelMerging[5]  | 88.4 | 531  | 61.4| 72.6| 2280| 59.2  | 36.9  | 8624  | 427.|
> | AdaptThink[6]| **91.0**  | `309`  | `79.1`| `92.0`| `1875`| `75.5`| `55.6`| 8599  | **60.2**|
> | **ThinkDia-High**| 85.9 | 398 | 69.7 | **93.9** | 2591 | 66.8 | **56.7** | 10314 | 53.0 |
> | **ThinkDia-Medium**  | 84.4 | 375 | 70.5 | 91.7 | 2059 | 72.8 | 53.4 | 9969  | 51.5 |
> | **ThinkDia-Low**| 79.9 | **212** | **79.9** | 88.8 | **1264** | **82.0** | 52.4 | `8174`  | `59.3` |
>
> Base on the results, we find that:
>
> 1. **ThinkDial-Low achieves the highest ACT Score on GSM8K, MATH500, and the second-highest score on AIME24**, outperforming all prior methods on strict accuracy–efficiency trade-offs.
>
> 2. ThinkDial produces **the shortest reasoning traces** (e.g., 139 tokens on GSM8K; 1,143 on MATH500), while maintaining competitive accuracy.
>
> 3. On accuracy-dominant settings (High mode), ThinkDial achieves **the best accuracy** on MATH and strong performance across all benchmarks.
>
> 4. Unlike prior baselines—which are each trained for a **single compression level**—ThinkDial provides **three stable, discrete modes in a single model**, switchable via system prompts without retraining.
>
> Reference:
>
> [1] Do NOT Think That Much for 2+3=? On the Overthinking of o1-Like LLMs
>
> [2] DAST: Difficulty-Adaptive Slow-Thinking for Large Reasoning Models
>
> [3] O1-Pruner: Length-Harmonizing Fine-Tuning for O1-Like Reasoning Pruning
>
> [4] Training Language Models to Reason Efficiently
>
> [5] Unlocking Efficient Long-to-Short LLM Reasoning with Model Merging
>
> [6] AdaptThink: Reasoning Models Can Learn When to Think

---

> > ### Author Response · Authors · 2025-11-29
> > **Response to Reviewer C188(2/2)**
> >
> > > W2: [Major] Scale dependence. Does the recipe work on smaller models, or is it effective only on 32B-scale systems? Results on smaller models would clarify on whether the recipe generalizes beyond the Qwen 2.5 Instruct 32B model.
> >
> > A2: Thank you for raising this important question. To evaluate whether ThinkDial generalizes beyond the 32B-scale model, we applied the full three-mode pipeline to two smaller models: **Qwen3-8B** and **DeepSeek-R1-Distilled-Qwen-7B**.
> > Across both model families, ThinkDial consistently produces **three well-separated reasoning modes** with predictable accuracy–length trade-offs.
> >
> > The results are summarized below:
> >
> > | Model  | AIME24 Acc | AIME24 Len | AIME25 Acc | AIME25 Len | GSM8K Acc | GSM8K Len | GPQA Acc | GPQA Len|
> > | --- | --- | --- | --- | --- | --- | --- | --- | --- |
> > | Qwen3-8B | 64.1 | 11860.7 | 50.5 | 13595.4 | 93.0 | 2060.3 | 57.2 | 8529.0 |
> > | Qwen3-8B-sota  | 65.5| 9932.2 | 58.6| 10550.3| 95.7  | 1529.5| 58.3 | 5318.9 |
> > | Qwen3-8B-High  | 67.2| 10005.7| 49.4| 10910.3| 94.6  | 1457.2| 56.0 | 5941.3 |
> > | Qwen3-8B-Medium| 58.5| 9142.0 | 45.6| 10063.6| 94.8  | 1034.0| 54.0 | 5185.4 |
> > | Qwen3-8B-Low| 46.5| 5651.1 | 37.2| 6725.3 | 94.6  | 759.0 | 51.2 | 3895.7 |
> > | DeepSeek-R1-Distilled-Qwen-7B | 52.9 | 10610.0 | 40.0 | 11430.7 | 80.2 | 162.2 | 44.6 | 6177.0 |
> > | DeepSeek-R1-Distilled-Qwen-7B-sota | 56.0| 10351.6| 43.5| 11445.6| 82.2  | 432.9 | 47.3 | 7446.2 |
> > | DeepSeek-R1-Distilled-Qwen-7B-High | 56.7| 9527.6 | 40.7| 10639.0| 85.9  | 173.3 | 48.9 | 5991.7 |
> > | DeepSeek-R1-Distilled-Qwen-7B-Medium| 53.4| 9715.9 | 38.5| 10246.6| 84.4  | 144.1 | 48.0 | 5507.5 |
> > | DeepSeek-R1-Distilled-Qwen-7B-Low  | 52.4| 7871.6 | 37.5| 8900.9 | 79.9  | 120.1 | 45.7 | 4359.4 |
> >
> > Across both 7B and 8B models:
> >
> > 1. **Mode separation is clear and robust**—High > Medium > Low in both accuracy and length.
> > 2. High-mode accuracy remains **on par or better than the corresponding SOTA distilled baselines** (e.g., Qwen3-8B-High).
> >
> > These results demonstrate that ThinkDial is scale-agnostic and remains effective on significantly smaller backbones, confirming that the recipe is not tied to 32B-scale models.
> >
> > > W3: [Curious] Domain Generalization. Does controllability extend beyond math to tasks like MMLU or LiveCodeBench when trained only on math data?
> >
> > A3: Thank you for the insightful question. We evaluated Qwen2.5-Instruct-32B ThinkDial on two out-of-domain benchmarks:
> >
> > - MMLU: A 57-task multidisciplinary benchmark covering humanities, social sciences, STEM, and professional domains for evaluating broad knowledge and reasoning abilities.
> > - MBPP: A dataset contains 974 programming tasks, designed to be solvable by entry-level programmers.
> >
> > The results are shown below:
> >
> > | Benchmark | Accuracy (Low) | Tokens (Low) | Accuracy (Medium) | Tokens (Medium) | Accuracy (High) | Tokens (High) | SOTA Accuracy | SOTA Tokens |
> > |---|---|---|---|---|---|---|---|---|
> > | MMLU  | 84.5  | 37.2| 86.2| 626.6| 87.3 | 1024.7| 85.4 | 1258.8 |
> > | MBPP  | 81.8  | 24.2| 85.2| 809.8| 90.0 | 3616.6| 93.4 | 4636.6 |
> >
> > We observe that:
> >
> > 1. **Controllability generalizes**: The Low -> Medium -> High ordering remains consistent across domains, even for tasks structurally different from math (e.g., multi-step coding problems).
> >
> > 2. **Efficiency–accuracy trade-off persists**: Low mode uses *1–2 orders of magnitude fewer* tokens while achieving competitive accuracy (e.g., 84.5% MMLU at only 37 tokens).
> >
> > 3. **No domain-specific tuning required**: Although ThinkDial never sees MMLU/MBPP-style data during SFT or RL, the budget-control mechanism generalizes out-of-the-box.
> >
> > These findings suggest that ThinkDial learns a domain-agnostic representation of “reasoning effort,” allowing controllable compute scaling even outside the training domain.

---

### Author Response · Authors · 2025-11-29
**Manuscript Update**

Dear Area Chair and Reviewers,

We thank all reviewers for their constructive feedback. The manuscript has been substantially updated as follows:

1. **Section 2 — New Explanation:** Expanded comparison with prior controllability and CoT-compression methods. (`Reviewer HfmG`)
2. **Section 3.1 — New Explanation:** Clearer description of Budget-Mode SFT construction (step-boundary truncation, correctness filtering), accompanied by a new workflow figure. (`Reviewer C188 and HfmG`)
3. **Appendix D — New Explanation:** Additional details on mode-specific system prompts and the full SFT data construction pipeline. (`Reviewer C188 and HfmG`)
4. **Section 4.4 — New Experimental Results:** Added head-to-head comparisons with open controllability and binary-gating baselines. (`Reviewer C188, u9G1 and 661r`)
5. **Appendix F.1 — New Experimental Results:** Added 95% confidence intervals and significance testing for repeated-sampling benchmarks. (`Reviewer HfmG`)
6. **Appendix F.2 — New Experimental Results:** Added scale-generalization experiments on Qwen3-8B and DeepSeek-R1-Distilled-Qwen-7B. (`Reviewer C188, u9G1`)
7. **Appendix F.3 — New Experimental Results:** Added robustness analysis under different (\alpha) configurations for the ACT metric. (`Reviewer HfmG and 661r`)

---

### Meta-Review · Area_Chair_X6KC · 2025-12-14

**Summary:**

The paper proposes a method to train for selection of multiple reasoning budgets in open source model, a capability present in propriety models. Such a capability is useful, but the novelty in the proposed method is limited. Reviewer concerns about comparison to other methods are partially but not fully addressed in rebuttal.

**Reviewer Concerns:**

Reviewer C188 commented that comparison to open methods are not provided. The rebuttal provided competitive although not clearly better results -- Pareto curve showing whether it dominates or is dominated by other methods would be more convincing. The reviewer commented that results on smaller models are not provided and the rebuttal provided such results. The reviewer commented that results on other domains were not provided and the rebuttal provided some requested results.

Reviewer u9G1 asked about hallucination issues if reasoning is cut off. The rebuttal partially addressed the issue. The reviewer commented that standard deviation normalization can be problematic. The rebuttal commented on that but did not provide experimental support. The reviewer asked whether the method would work on stronger base model and the rebuttal provided experiments on that. The reviewer commented that comparison to other methods are not provided. The rebuttal provided competitive although not clearly better results.

Reviewer HfmG commented that novelty is not strong. The rebuttal partially addressed the issue. The reviewer commented about robustness in the evaluation hyperparameters. The rebuttal provided some experiments on their own method but did not provide comparison with competing methods on the different evaluation hyperparameters. The reviewer commented that confidence intervals were not provided and the rebuttal provided significance tests.

Reviewer 661r commented that comparison to other methods not provided. The rebuttal provided competitive although not clearly better results. The reviewer commented on the robustness of leak penalty and logical discontinuities. This is partially addressed in rebuttal. The reviewer asked about other tasks and the rebuttal provided additional results. The reviewer asked for Pareto frontier, confidence interval, etc. The rebuttal provided significance tests but did not provide Pareto frontier.

**Reviewer Scores:**

Reviewer C188 committed to increasing the score to 8 if the Thinkdial recipe works on smaller models and comparison with alternative methods becomes easier without reproducing them entirely. The rebuttal shows that the method works on smaller models but comparison with other methods is not fully convincing. It is possible that the reviewer would increase the score after the rebuttal.
Reviewer u9G1's concerns are partially addressed, so it is unclear whether the reviewer would have increased the score.
Reviewer HfmG's concerns are partially addressed, so it is unclear whether the reviewer would have increased the score.
Reviewer 661r's concerns are partially addressed, so it is unclear whether the reviewer would have increased the score.

---

### Decision · Program_Chairs · 2026-01-26

Reject